# VTA dopamine neurons are hyperexcitable in 3xTg-AD mice due to casein kinase 2-dependent SK channel dysfunction

Harris E. Blankenship [1,2], Kelsey A. Carter[1], Kevin D. Pham[3], Nina T. Cassidy[1,3], Andrea N. Markiewicz[1], Michael I. Thellmann [1], Amanda L. Sharpe [4], Willard M. Freeman [3,5] & Michael J. Beckstead [1,2,5] ✉

Alzheimer's disease (AD) patients exhibit neuropsychiatric symptoms that extend beyond classical cognitive deficits, suggesting involvement of subcortical areas. Here, we investigated the role of midbrain dopamine (DA) neurons in AD using the amyloid + tau-driven 3xTg-AD mouse model. We found deficits in reward-based operant learning in AD mice, suggesting possible VTA DA neuron dysregulation. Physiological assessment revealed hyperexcitability and disrupted firing in DA neurons caused by reduced activity of small-conductance calcium-activated potassium (SK) channels. RNA sequencing from contents of single patch-clamped DA neurons (Patch-seq) identified up-regulation of the SK channel modulator casein kinase 2 (CK2), which we corroborated by immunohistochemical protein analysis. Pharmacological inhibition of CK2 restored SK channel activity and normal firing patterns in 3xTg-AD mice. These findings identify a mechanism of ion channel dysregulation in VTA DA neurons that could contribute to behavioral abnormalities in AD, paving the way for novel treatment strategies.

Ninety-eight percent of Alzheimer's disease (AD) patients experience neuropsychiatric symptoms ranging from anxiety to clinical depression[1]. These conditions cannot be fully accounted for by the cortical and hippocampal degeneration typically associated with AD and are consistent with subcortical dysregulation. Recent findings from preclinical AD models include fewer tyrosine hydroxylase positive (TH+) neurons commensurate with decreased dopamine (DA) release[2], aberrant DA neuron firing patterns[3,4], and increased DA receptor expression, particularly in the ventral tegmental area (VTA)[5]. However, investigations into single-neuron activity and changes in cellular physiology during disease progression are sparse[3,4,6–9] and predominantly limited to studies of cortical and hippocampal pathophysiology in amyloid-only models during prodromal stages of the disease.

VTA dopaminergic neuron projections to the cortex contribute to episodic memory formation[10–12] while subcortical projections, particularly to the nucleus accumbens (NAc), encode reward, motivation, and aversion[13,14]. While tests of cognition[2] and locomotion[5] are common in the preclinical AD literature, reward learning is largely unexplored. Reward-based behaviors rely on both tonic extracellular DA regulated by basal firing of dopaminergic neurons and phasic release of DA mediated by burst firing[15–17]. In the substantia nigra pars compacta (SNc), autonomous firing of DA neurons is central to the initiation of degeneration in Parkinson's disease (PD)[18–22]. Conversely, in AD, the mesocorticolimbic system (i.e., the VTA) seems to be preferentially affected over the nigrostriatal pathway[2,23,24]. Distinct biophysical mechanisms underlie firing patterns in the SNc and VTA[16,25], potentially contributing to the selectivity of neurodegenerative diseases within

[1]Aging and Metabolism Research Program, Oklahoma Medical Research Foundation, Oklahoma City, OK, USA. [2]Department of Physiology, University of Oklahoma Health Sciences Center, Oklahoma City, OK, USA. [3]Genes and Human Disease Research Program, Oklahoma Medical Research Foundation, Oklahoma City, OK, USA. [4]Department of Pharmaceutical Sciences, University of Oklahoma Health Sciences Center, Oklahoma City, OK, USA. [5]Oklahoma City Veterans Affairs Medical Center, Oklahoma City, OK, USA. ✉e-mail: mike-beckstead@omrf.org

DA neuron subpopulations. Ion channels govern the electrophysiological properties of excitable cells, and hippocampal neurons exhibit aberrant ion channel function and action potential generation in AD models[26,27]. As single neurons are the computational units of the brain, determining physiological alterations of individual cell populations in early AD is central for understanding disease pathophysiology and for identifying potential therapeutic targets.

Here, using the amyloid + tau-based triple-transgenic 3xTg-AD mouse model (3xTg)[28,29], we describe deficits in reward-based learning that prompted an investigation into firing and ion channel function in single VTA DA neurons. We observed hyperexcitability of DA neurons caused by decreased small-conductance calcium-activated potassium (SK) channel currents in 3xTg compared to age-matched control mice. This was driven by SK-bound casein kinase 2 (CK2), which decreases the calcium binding affinity of calmodulin (CaM) and effectively reduces SK channel activation by calcium. Importantly, pharmacological inhibition of CK2 was sufficient to restore basal firing rate and regularity in 3xTg DA neurons. These results reveal a noncanonical deficit in dopaminergic behavior and single cell physiology that hinges on hyperactivity of a single AD-associated enzyme, CK2, implicating it as a potential therapeutic target for AD.

## Results

### Reinforcement learning is impaired in 3xTg-AD mice

3xTg mice carry homozygous transgenes for the human amyloid precursor protein mutant $APP_{Swe}$, $tau_{P301L}$, and $PS1_{M146V}$, all of which are linked to familial neurodegeneration. These mice accumulate both extracellular beta-amyloid (Aβ) and intracellular hyperphosphorylated tau[28] in a stereotyped manner. Cognitive learning deficits begin to emerge in 3xTg mice at four to six months of age[30]. In the 3xTg hippocampus, Aβ plaques are present by six months[29], and hyperphosphorylated tau is detectable by twelve months of age[29] (https://modeladexplorer.org/). This stereotyped pathological progression provides the framework for explorations of neuronal dysfunction in an established model of familial AD.

AD patients suffer from deficits in motivation-based appetitive behaviors that could indicate a central role for VTA DA neurons[23]. To determine whether 3xTg mice display behavioral impairment in operant reward learning, we assessed acquisition of an operant task to gain access to palatable chocolate-flavored pellets in non-food restricted 3-, 6-, and 12-month-old 3xTg and wildtype (WT) mice (Fig. 1a), a task previously shown to rely on NAc DA[31-33]. All WT mice tested at all three ages successfully acquired the operant task. However, 3xTg mice at 6- and 12-months exhibited slower learning of the task, and several mice were unable to learn the task at all, especially at the 12-month time point (Fig. 1b). Raster plots of nose poke responding on day 1 of training for individual mice that eventually learned the task show less responding over the 3-h session compared to WT mice, although some 3xTg mice did exhibit responding similar to WT mice (Fig. 1c). Among the mice who acquired the task, there was no main effect of age or genotype on the mean number of pellets earned during a session (Supplementary Fig. 1a) or on motivation to gain access to the reinforcer, as measured by progressive ratio breakpoint (Supplementary Fig. 1b). One potential explanation for impaired instrumental learning would be if 3xTg mice exhibited decreased locomotor activity. However, in line with previous reports[5,34], we detected increased locomotion in 3- and 12-month-old 3xTg mice (Fig. 1d–g) assayed by total distance traveled (Fig. 1h). Additionally, 3xTg mice also spent significantly more time in the center of the locomotor area, which could suggest an increase in exploratory behavior or a decrease in anxiety-like behavior (Fig. 1i)[35]. To further investigate these clear deficits in reward learning[31] in 3xTg mice, we next aimed to probe the physiological properties of single DA-releasing neurons in the VTA.

### 3xTg DA neurons exhibit increased firing rate and irregularity

While DA release can occur independent of somatic activity[36,37], alterations in intrinsic DA neuron physiology are sufficient to affect reinforcement learning[38]. To examine the physiological properties of VTA DA neurons across the lifespan and in response to amyloid and tau expression, patch clamp recordings were made in brain slices from WT and 3xTg mice at 3, 6, 12, and 18 months of age. In rodent brain slices, VTA DA neurons spontaneously fire in a slow (0.5–4 Hz), pacemaker-like pattern[39]. Using the cell attached configuration (Fig. 2a) to assess spontaneous firing without disrupting intracellular contents, we observed increased spontaneous firing frequency in 3xTg mice compared to age-matched controls (Fig. 2b). 3xTg mice also displayed lower firing rhythmicity as indicated by an elevated coefficient of variation (CV) of inter-spike intervals (ISI; Fig. 2c). While the CV of the ISI was high throughout the lifespan, the increase in firing rate exhibited an inverted-U shape, peaking at 12 months before declining to WT frequency at 18 months (Fig. 2b). Since the ionic mechanisms that control pacemaking overlap with those governing a shift to burst firing VTA DA neurons[15,40], we next sought to determine how 3xTg neurons responded to electrical input.

In vivo, glutamatergic afferents drive VTA DA neuron burst firing in response to reward-predictive cues[41]. These synaptic connections are severed in ex vivo brain slice preparations, and DA neurons do not typically burst under basal conditions. Somatic current injection can mimic burst activity and provide insight into how DA neurons respond to excitatory input[15]. Using whole-cell current-clamp recordings, we next applied a series of current steps (0–250 pA) to generate frequency–current ($F$–$I$) curves for WT and 3xTg mice of all four ages (Fig. 2d, e). 3xTg DA neurons were hypersensitive to current injection, as quantified by the slope of F-I curves, or gain (Fig. 2f). To further assess the effects on firing, we employed the minimally-invasive gramicidin perforated patch recording technique to produce stable, long-duration recordings while maintaining electrical access to the cell. Focusing on 12-month-old mice, we again observed a robust increase in frequency and irregularity in pacemaking 3xTg DA neurons (Fig. 2g–i). Additionally, in this recording configuration, we were able to detect a significant depolarization of the apparent spike threshold in spontaneously firing 3xTg DA neurons (Fig. 2j), but no change in the spike width (WT: 1.488 ± 0.0025 ms, 3xTg: 1.488 ± 0.0040 ms, P = 0.9781), suggesting high threshold calcium channels and action potential repolarization mechanisms are unaltered[42]. To test whether altered firing generalized to other catecholaminergic neurons, we also recorded from two additional populations: locus coeruleus (LC) norepinephrine (NE)-releasing neurons (Supplementary Fig. 2) and SNc DA neurons. Both of these neuron types exhibit spontaneous firing activity that is governed by similar ion channel conductances to those of VTA neurons[43], in addition to the LC being the site of the earliest observed hyperphosphorylated "pretangle" tau in humans[44]. While a recent study showed subtle in vivo firing pattern changes in LC NE neurons of mutant tau expressing mice[45], we did not detect changes in spontaneous firing rate (Supplementary Fig. 2a, b), firing fidelity (Supplementary Fig. 2c), or somatically-evoked firing (Supplementary Fig. 2d–f) in LC NE neurons from 12-month-old 3xTg mice. In the SNc, we also detected no alterations in spontaneous firing rate or CV of ISI in 12-month-old 3xTg mice (Supplementary Fig. 3a–d). Furthermore, evoked firing in 3xTg mice (Supplementary Fig. 3e–g) was similar to that of WT mice. These findings are consistent with a prior report of preserved SNc DA neuron function in an amyloid-only model of AD[2]. Thus, VTA DA neurons appear to be uniquely susceptible among multiple populations of catecholaminergic neurons in 12-month-old 3xTg mice.

The biophysical properties responsible for slow spontaneous firing of midbrain DA neurons are extremely well characterized. In the VTA, the two major outward currents are carried by A-type potassium (Kv4.3)[46-48] and SK (largely SK3) channels[40,49-51], which interact across

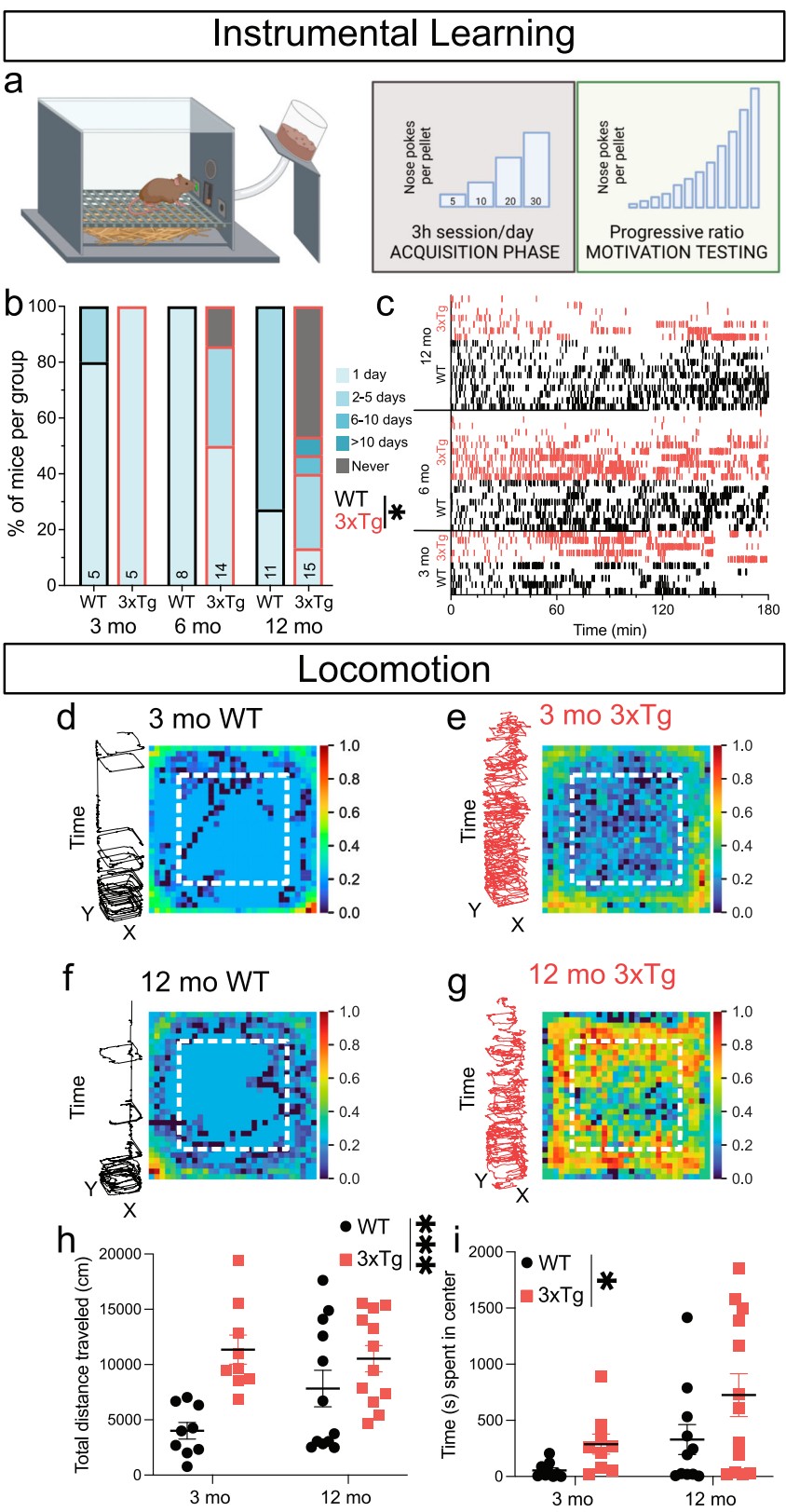

the ISI to maintain a stable firing cycle. We sought to determine how ISI dynamics in VTA DA neurons differ between WT and 3xTg mice. Interspike voltage trajectories were collected and averaged for both genotypes (Fig. 2k). The mean medium afterhyperpolarization (mAHP) voltage was depolarized in DA neurons from 3xTg mice (Fig. 2l) as was the minimum voltage of the ISI (Fig. 2m). The mAHP is governed largely by action potential mediated influx of calcium and subsequent

activation of SK channels[52]. Indeed, the reduced mean mAHP and minimum ISI voltages in 3xTg mice were highly reminiscent of data we recently described in the SNc following a partial block of SK channels using apamin[53], and a depolarized threshold may be secondarily linked to SK channels through sodium channel inactivation[50,54]. Thus, we next aimed to determine whether SK channel activity is altered in 3xTg VTA DA neurons.

**Fig. 1 | Instrumental reward learning is age-dependently impaired in 3xTg mice.**
**a** Schematic of procedure used for operant self-administration of palatable food pellets. **b** 3xTg mice acquired operant self-administration of palatable pellets more slowly than WT mice (main effect, $P = 0.0106$, $F_{1,52} = 7.034$) with the greatest effect between genotypes at 12-mo (two-tailed Sidak's multiple comparison, $P = 0.0003$). **c** Raster plots showing correct side nose poke responding (vertical ticks along the x-axis) on day 1 of training for individual mice (by row) that eventually acquired self-administration. While there was considerable variation within the 3xTg groups, overall they showed less responding on day 1 compared to WT mice.
**d–g** Representative locomotor data from individual 3- and 12-mo WT and 3xTg mice during the first thirty minutes in the chamber, stratified by time (tornado

plots) and heatmap localization (residency plots). **h** 3xTg mice locomote significantly more than WT mice over the 1-h session (two-way ANOVA, main effect of genotype, $P = 0.0006$, $F_{1,38} = 13.84$, $n = 42$ mice) which was most dramatic at 3 mo ($P = 0.0018$, two-tailed Sidak's). **i** 3xTg mice spent significantly more time in the center of the chamber during the 1-h session than WT mice (two-way ANOVA, main effect of genotype, $P = 0.0365$, $F = 1,38 = 4.699$, same animals as in **h**). Residency plot scalebars represent log-transformed data rescaled between minimum and maximum values. Error bars represent standard error. Source data are provided as a Source Data file. Created in BioRender. Beckstead, M. (2023) BioRender.com/p21u763.

## SK channel dysfunction underlies aberrant firing

SK channels are activated by calcium influx through voltage-gated channels and subsequent calcium-induced calcium release[4,55,56]. SK activity can be measured as an afterhyperpolarization (tail) current in SNc and VTA DA neurons following a depolarizing voltage step[54,57,58]. A decrease in this tail current has been previously reported in VTA DA neurons from 6-month-old Tg2576 mice (which have the APP$_{Swe}$ mutation)[4]. To first confirm that the tail current is generated by the SK channel, we applied a saturating concentration of the selective SK channel inhibitor apamin (100 nM) to WT VTA DA neurons (Supplementary Fig. 4). Apamin rapidly and consistently decreased the amplitude of the tail current from an average of 329.4 to 109.8 pA (Supplementary Fig. 4b, c). We therefore assessed tail currents using the same protocol (100 ms depolarizing step from −72 to −17 mV) in DA neurons from 3-, 6-, 12-, and 18-month-old WT and 3xTg mice (Fig. 3a). We observed markedly smaller tail current amplitudes (Fig. 3b) and area under the curve (AUC; Fig. 3c) in 3xTg versus WT mice. There was also a significant main effect of age on both measures (Fig. 3b, c). In contrast, we did not detect any change in the tail current AUC or maximal amplitude in LC NE neurons (Supplementary Fig. 2g–i) or SNc DA neurons (Supplementary Fig. 3h, i) from 12-month-old 3xTg mice, again demonstrating selectivity for VTA DA among catecholaminergic neuron populations.

To determine whether partial block of SK channels recapitulates the physiological abnormalities seen in 3xTg mice, we applied a low concentration of apamin (1−3 nM) to spontaneously firing VTA DA neurons from 12-month-old WT mice in the perforated patch configuration (Fig. 3d). We observed an increase in firing rate (Fig. 3e) and CV of ISI (Fig. 3f) similar to what is observed in 3xTg neurons. Moreover, the ISI dynamics of spontaneously firing VTA DA neurons mimicked those from 3xTg mice (Fig. 3h), with mean mAHP voltage (Fig. 3i), minimum ISI voltage (Fig. 3j), and spike threshold (Fig. 3g) all depolarized. These data are consistent with SK channel involvement in the physiological abnormalities of VTA DA neurons in 3xTg mice.

SK channel dysfunction in VTA DA neurons has previously been linked to deficits in attention gating, possibly due to reduced effects on NMDA receptor-mediated currents[58]. We therefore sought to determine if similar behavioral abnormalities could be observed in 3xTg mice (Supplementary Fig. 5a). Using a Pavlovian attention task, both 3xTg and WT mice were able to learn to discriminate between cue tones that predicted a high (100%) and low (12%) probability for food reward, as indicated by a decreased latency to enter the food receptacle following the high-probability tone compared to the low-probability tone (Supplementary Fig. 5b). On a subsequent test day, the trials were randomly alternated between the high-probability tone and a compound stimulus consisting of a flashing light (novel stimulus) concurrently presented with the high-probability tone. In response to the novel stimulus distractor, all mice in early trials increase the latency to enter the food receptacle compared to the high-probability tone alone, and over trials shifted their attention away from the novel stimulus to retrieve the pellet in the same time as the high-probability tone alone. The WT and 3xTg mice performed similarly in response to the novel stimulus, suggesting no apparent deficit in attention gating

in 3xTg mice (Supplementary Fig. 5c). We next tested whether SK channel modulation of NMDA receptor currents was affected in 3xTg mice[58]. In whole-cell voltage clamp, we evoked NMDA receptor-mediated excitatory post synaptic currents (EPSCs) in brain slices from 12-month-old mice and applied the SK channel inhibitor apamin. Consistent with the prior report, apamin increased NMDA receptor-mediated currents in WT DA neurons (Supplementary Fig. 6a, b). Additionally, the effect of apamin was absent in cells from 3xTg mice (Supplementary Fig. 6c), consistent with dysregulation of SK channels in 3xTg DA neurons.

It's unclear why established effects of SK channel dysfunction in individual VTA neurons do not translate to deficits in attention gating in 3xTg mice. One possibility is that 3xTg mice may exhibit Wallerian-like degeneration, a process where axons die-back prior to neuron death. A previous report suggested denervation may occur in the ventral striatum of Tg2576 mice[2], using TH and dopamine transporter (DAT) expression as proxies for dopaminergic axons in the region. Using a similar immunohistochemical approach, we observed a significant decrease in TH and DAT staining in the ventral striatum of 3xTg mice compared to WT at 12 months of age (Supplementary Fig. 7a–c). Interestingly, we also detected a decrease in TH staining in the VTA, but no decline in the total number of TH$^+$ cells (Supplementary Fig. 7d–f), in contrast to reports from the Tg2576 amyloid-only model[2]. These results suggest DA neurons may lose neurites prior to cell death in the VTA of 3xTg mice, potentially minimizing the direct link between somatic hyperexcitability and DA output.

Beyond SK, deficits in other potassium channels in DA neurons can also result in hyperactivity and altered instrumental conditioning, including Kv4.3/A-type potassium channels ($I_A$)[38]. $I_A$ limits the rate of pacemaker firing in VTA DA neurons by slowing depolarization across the ISI[46,47,59]. To test for possible alterations in 3xTg mice, we used whole-cell voltage-clamp recordings to isolate $I_A$ using a series of voltage steps (Supplementary Fig. 8a)[47,48,59]. Neither $I_A$ amplitude (Supplementary Fig. 8b) nor decay time constant (Supplementary Fig. 8c) differed between WT and 3xTg VTA DA neurons, nor was there a difference observed in SNc DA neurons (Supplementary Fig. 3j, k). This suggests $I_A$ is unaltered in midbrain DA neurons of 3xTg mice.

## 3xTg VTA DA neurons display decreased sensitivity to NS309

Positive allosteric modulators of SK channels have previously been shown to safeguard neurons from neurotoxic insults in degenerative models[60,61]. We next tested the sensitivity of VTA DA neurons by applying the selective SK channel positive allosteric modulator NS309 (1 μM) in perforated-patch recordings of WT (Fig. 4a) and 3xTg VTA DA neurons (Fig. 4b)[53,62]. NS309 reduced firing rate in WT (−35.5% ± 13.5%), but not 3xTg neurons (−10.7% ± 16.2%, Fig. 4c). However, it had no significant impact on the CV of the ISI in either genotype (Fig. 4d), contrary to our recent findings in SNc neurons[53]. Further analysis of ISI dynamics revealed that voltage trajectories in WT neurons (Fig. 4e) appeared to be more sensitive than 3xTg neurons (Fig. 4f). NS309 significantly hyperpolarized the average mAHP (Fig. 4g) and the minimum ISI voltage (Fig. 4h) in WT but not 3xTg VTA DA neurons.

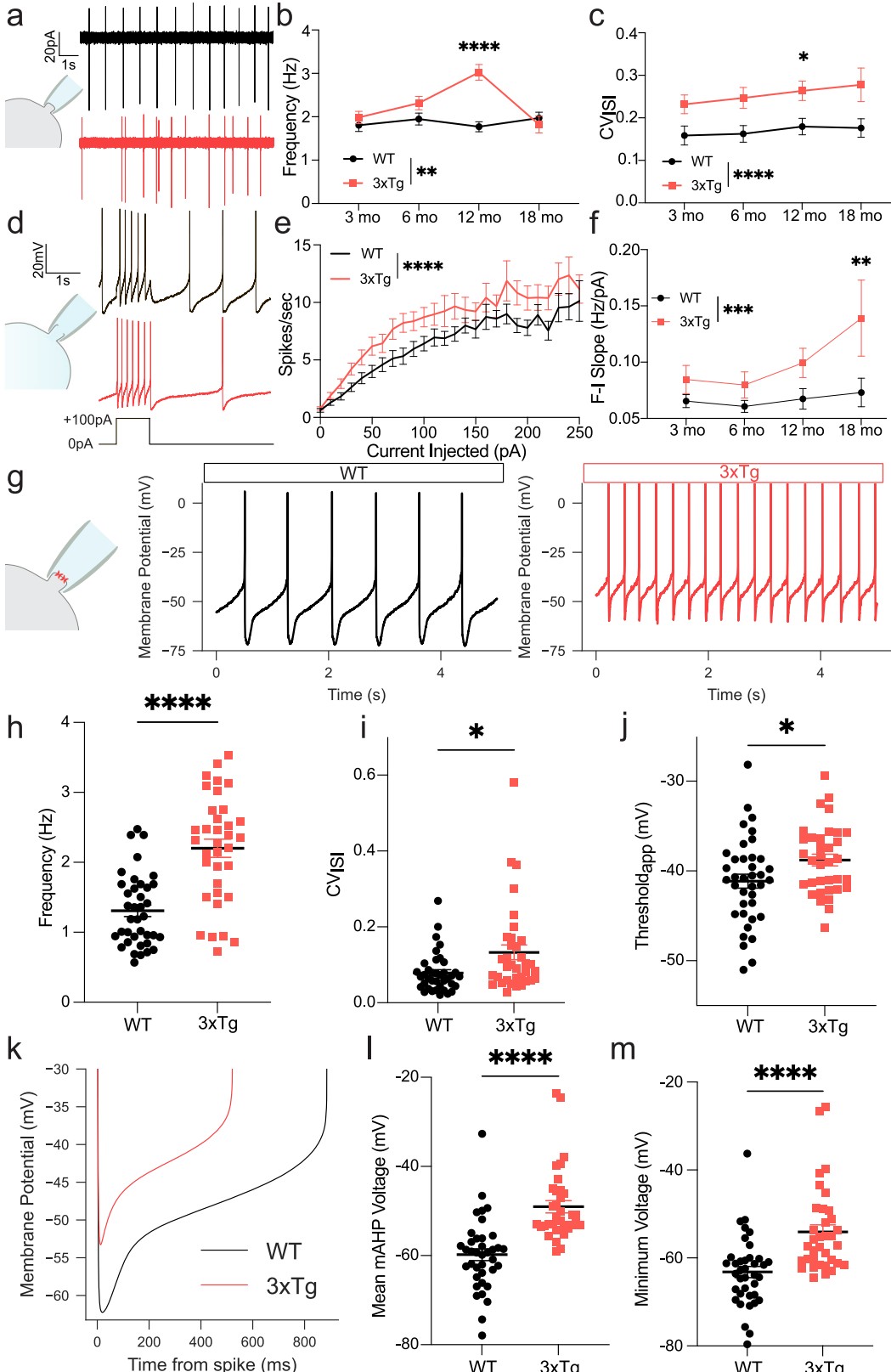

These data suggest that SK channels in 3xTg mice might be less sensitive to positive allosteric modulation by NS309.

SK channels function as multi-protein complexes that continuously associate with CaM as the calcium sensor[63]. $Ca^{2+}$ binding to CaM produces a conformational change in the channel that results in pore opening and potassium efflux[64]. NS309 augments the apparent calcium sensitivity of SK channels by binding to the channel-CaM interface, allosterically enhancing phosphatidylinositol 4,5-bisphosphate (PIP2) recruitment to promote channel opening[65,66]. SK channel-bound CaM can be dynamically phosphorylated at threonine 79[67], and upon phosphorylation the NS309 binding affinity is decreased[66]. Our findings point toward the possibility that an

**Fig. 2 | 3xTg DA neurons display increased firing rate with decreased regularity.**
**a** Cell-attached recording and 10 s representative traces of spontaneous firing from 12 mo WT (black) and 3xTg (red) mice. **b** Mean spontaneous firing rate for WT and 3xTg DA neurons (cells/mice, WT: 3 mo 64/12, 6 mo: 64/14, 12 mo 76/17, 18 mo 43/7; 3xTg: 3 mo 75/6, 6 mo 73/8, 12 mo 88/22, 18 mo 35/6; two-way ANOVA $F_{genotype}$ = 9.236, $P$ = 0.0025; $F_{age}$ = 3.878, $P$ = 0.0092; $F_{interaction}$ = 5.943, $P$ = 0.0005), with the largest rate increase at 12 mo (two-tailed Sidak's, $P < 0.0001$). **c** Mean coefficient of variation of the ISI ($CV_{ISI}$) for the same cells (two-way ANOVA, $F_{genotype}$ = 19.64, $P < 0.0001$) were different between 3xTg and WT at 12 mo (two-tailed Sidak's, $P$ = 0.0214). **d** Whole-cell recording and representative traces of 100 pA, 1 s current injection in WT (black) and 3xTg (red) mice. **e** Averaged frequency/current (F/I) curves from 12 mo WT and 3xTg mice (two-way ANOVA, WT = 24 cells, 10 mice; 3xTg = 31 cells, 8 mice; $F_{genotype}$ = 54.56; $P < 0.0001$, $F_{current\ injection}$ = 18.48, $P < 0.0001$). **f** Hypersensitivity of 3xTg DA neurons measured as the F/I slope from 0 to 60 pA (cells/mice, WT: 3 mo 36/5, 6 mo 35/8, 12 mo 24/6, 18 mo: 34/7; 3xTg: 3 mo 35/3, 6 mo 17/5, 12 mo 32/7, 18 mo 17/3; $F_{genotype}$ = 13.71, $P$ = 0.0003; $F_{age}$ = 2.692, $P$ = 0.0471), with a significant increase in sensitivity at 18 mo in 3xTg DA neurons (two-tailed Sidak's, $P$ = 0.0044). **g** Gramicidin perforated-patch recording and representative traces from WT (black) and 3xTg (red) mice. **h** 3xTg neurons showed increased firing frequency (WT 38/10, 3xTg 35/8; two-tailed Mann–Whitney, $P < 0.0001$, $U$ = 240), and (**i**) increased $CV_{ISI}$ (two-tailed Mann–Whitney, $P$ = 0.0114, $U$ = 437). **j** Depolarized apparent threshold in 3xTg DA neurons (two-tailed $t$-test, $P$ = 0.0247, $t$ = 2.295). **k** Grand averages of ISI voltage trajectories for WT and 3xTg cells. **l** Depolarized mean medium after-hyperpolarization voltage (mAHP, two-tailed Mann–Whitney, $P < 0.0001$, $U$ = 168), and **m**. minimum ISI voltage (two-tailed Mann–Whitney, $P < 0.0001$, $U$ = 168) in 3xTg DA neurons. Cells in (**i**–**m**) are the same as in (**h**). Error bars represent standard error. Raw data provided as a Source Data file.

additional subcellular participant might phosphorylate CaM to alter basal and modulated SK channel activity in 3xTg DA neurons. To investigate the expression of SK channel interactors in DA neurons from WT and 3xTg mice, we turned to a molecular screening technique.

## Patch-seq of VTA DA neurons

To identify endogenous SK channel interactors, we coupled patch clamp electrophysiology with single-cell transcriptomics (Patch-seq)[68]. This technique was originally developed to profile neuronal subpopulations[68,69]; however, we posited it could be used as a screening assay akin to single-cell or single-nucleus RNA sequencing. The latter techniques have proven instrumental in detecting genes responsible for disease vulnerability in both post-mortem human brain tissue and animal models of disease[70,71]. Patch-seq has been used previously in this context to study pancreatic β-cells in a model of type-2 diabetes[72], in an in vitro model of schizophrenia[73], and in a model of corticospinal tract degeneration[74]. In each case, potential mechanistic targets of physiological dysfunction were identified, but this technique has not been previously used in brain sections from a preclinical disease model (including those for Alzheimer's or other neurodegenerative diseases).

Using slightly modified procedures to minimize artifacts and cell swelling[75], we obtained patch clamp recordings from 12-month-old WT and 3xTg mice (Fig. 5a). We again observed reduced SK currents in 3xTg mice and extracted a total of 68 cells from the two groups after quality control (QC) filtering for cells that expressed DA markers and did not demonstrate contamination with markers from other cell types (Fig. 5b). Initial analysis indicated consistent count depth, identification of an average 3000+ genes, and robust coverage across the detected genes (Supplementary Fig. 9a–c). Four thousand one hundred and sixty-five genes passed criteria (read count >1 in >80% of samples from at least one group) for expression. Expressed transcriptome profiles for all WT and 3xTg cells were visualized by t-distributed stochastic neighbor embedding (t-SNE; Fig. 5c). Using thresholds of FDR <0.1 and a |fold change| >1.25, there were 427 differentially expressed genes (DEGs), most upregulated (376 higher and 51 lower in 3xTg than in WT; Fig. 5d). DEGs were then assessed for Pearson correlation with individual cell tail current maximum amplitude (Fig. 5e, Supplemental Table 1). Gene Ontology analysis from DEGs indicates glycolytic and response to inflammatory stimuli as top hits, with increased glutamatergic synaptic markers in the top ten (Fig. 5f). From the DEGs, example transcripts of interest were identified (Fig. 5g). *Csnk2a1*, which codes for CK2, and the direct CK2 phosphorylation target *Nucks1* were both upregulated. *Slc25a14*, which encodes the mitochondrial uncoupling protein 5 was upregulated while *Nacc2*, which codes for NAc-associated protein 2 was significantly downregulated. Of the genes of interest, the one that provided the greatest clue into SK channel dysregulation was *Csnk2a1*, whose expression was negatively correlated with tail current

amplitude (Fig. 5e). CK2 is a ubiquitous kinase and has been a focus of the cancer metastasis field for decades[76]. However, CK2 also has a critical central nervous system role as a kinase that binds to SK channels and regulates their calcium sensitivity[63,67]. To confirm that changes in *Csnk2a1* transcript were relevant to protein levels, we performed brain slice immunofluorescence against CK2, phospho-calmodulin (p-CaM), and SK3 channels in the VTA. To assay changes specifically in DA neurons and not intermingled cell types, we co-labeled the DA neuron marker TH. We used TH staining to segment neurons and calculated the intensities of each of the three targets confined specifically to DA cell bodies (Fig. 6a). CK2 expression was slightly elevated in the 3xTg group (Fig. 6b, c) while its target, p-CaM showed a stronger rightward shift in the cumulative probability curve (Fig. 6d, e). Finally, SK3 levels were strongly rightward shifted (Fig. 6f, g), indicating an elevation in SK3 protein expression in 3xTg VTA DA neurons. This indicated that SK channel expression itself is not the underlying mechanism of SK dysfunction in 3xTg mice. We thus sought to examine the role of CK2 in SK channel regulation in WT and 3xTg mice.

## CK2 is central to DA neuron dysfunction in 3xTg mice

CK2 modulates SK channels in an activity-dependent manner[67,77,78], and previous work has highlighted SK-associated CK2 activity in a model of *status epilepticus*[79], another form of hyperactivity. CK2 is upregulated in AD patient brain tissue[80,81] and animal models of the disease[82]. CK2 hyperactivity has also been previously associated with cognitive decline in 3xTg mice[83]. Further, Aβ stimulates CK2 activity in vitro[83,84] providing a potential mechanistic link between AD pathology and dopaminergic dysfunction.

To test the effects of CK2 phosphorylation on SK channel currents, we incubated brain slices from 12-month-old WT and 3xTg mice for 1–3 h with the membrane-permeant CK2-specific antagonist SGC-CK-2 (1 μM)[85], and compared them to WT cells (previously presented in Fig. 2). SGC significantly reduced the spontaneous firing rate in 3xTg but not WT VTA DA neurons (Fig. 7a–c). The CV of ISI was also substantially reduced only in 3xTg and not WT neurons (Fig. 7d). Voltage trajectories were similarly affected (Fig. 7e); SGC had no apparent effect on the mAHP in WT cells but hyperpolarized 3xTg neurons to near WT levels (Fig. 7f). Similar effects were seen for the minimum ISI voltage (Supplementary Table 4, $P < 0.0001$). Additionally, apparent spike threshold was significantly hyperpolarized in SGC-treated 3xTg neurons (from -38.79 ± 0.645 to -42.49 ± 1.45 mV, $P$ = 0.0415) suggesting that physiological adaptations secondary to SK channel dysfunction[54] can be restored. We next compared tail current AUC in the presence of SGC to data presented in Fig. 3c (12-month timepoint). Consistent with our spontaneous firing data, we detected no effect of SGC on WT neurons, but a significant increase in tail current charge in 3xTg neurons (Supplementary Fig. 10a, b). To provide an overview of electrophysiological alterations across experimental groups, we included a statistical table of intrinsic property differences between WT and 3xTg neurons (Supplementary Table 2) as well as genotype

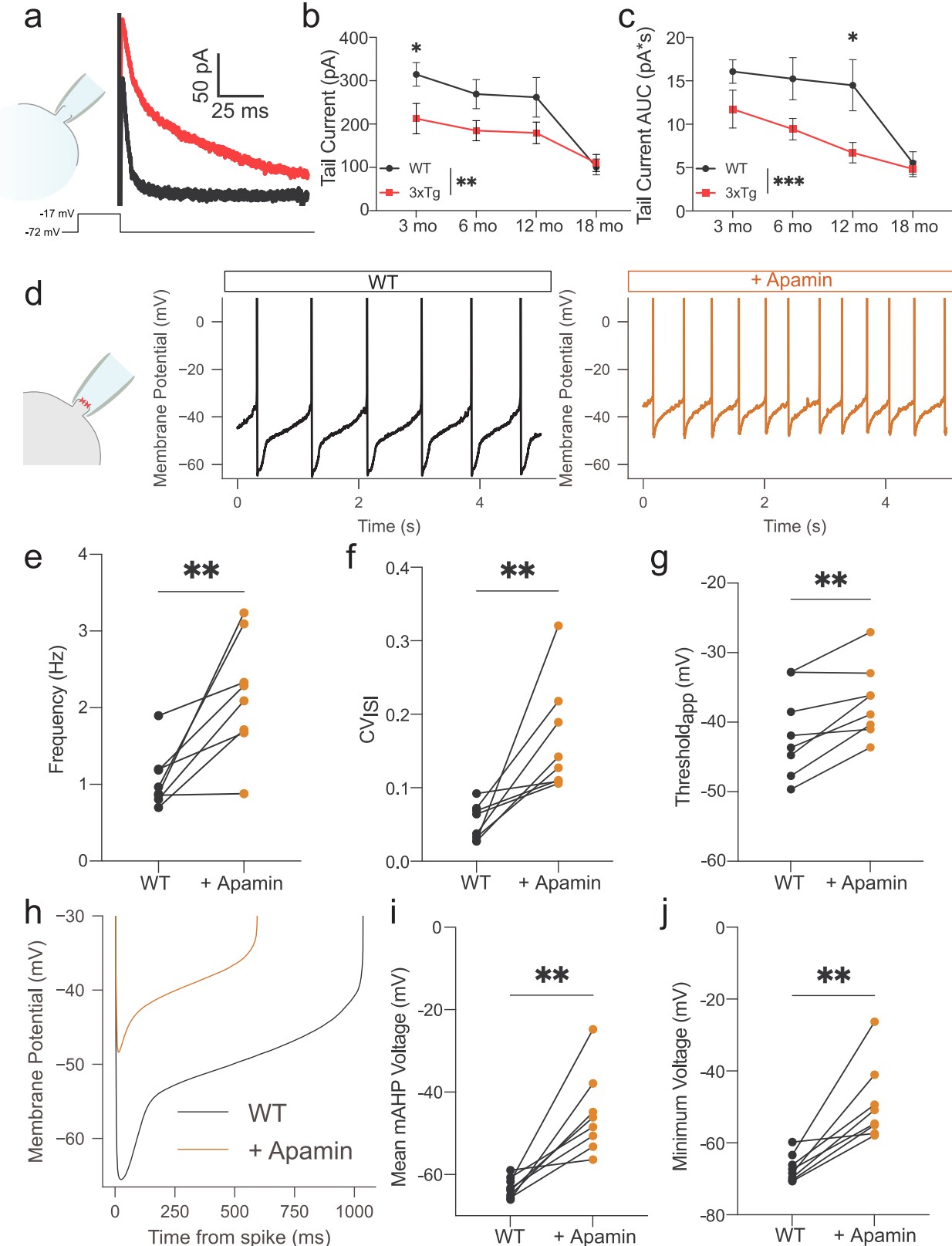

and pharmacological interventions and their effect on firing rate (Supplementary Table 3).

While CK2 inhibitors display high efficacy in both pre-clinical models[79,86,87] and clinical trials (ClinicalTrials.gov: NCT03897036; NCT0390486), to date SGC has been exclusively tested in vitro. To assess its effects in vivo, we treated 3xTg mice with either vehicle or SGC (1 mg/kg, i.p.). Mice received four days of pre-treatment which

consisted of twice-daily injections, followed by an experimental phase which consisted of a single injection, and an abstinence phase where mice were no longer injected for up to seven days prior to perforated patch electrophysiology experiments (Fig. 7g). Intraperitoneal injections of SGC reduced firing rate compared to vehicle-treated mice (Fig. 7h), an effect that trended towards significance after mice had not received injections for at least five days (Fig. 7i). Mean mAHP

**Fig. 3 | SK channel inhibition in WT neurons recapitulates the 3xTg phenotype.** **a** Representative traces of the tail current in response to a 100 ms depolarization step to −17 mV from $V_{hold}$ (−72 mV) in the whole-cell configuration. **b** Reduced tail current maximal amplitude (cells/mice, WT: 3 mo 66/7, 6 mo 25/7, 12 mo 20/6, 18 mo 37/6; 3xTg: 3 mo 42/3, 6 mo 40/8, 12 mo 42/7, 18 mo 28/6; two-way ANOVA $F_{genotype} = 8.743$, $P = 0.0034$; $F_{age} = 10.57$, $P < 0.0001$) and (**c**) area under the curve (AUC, $F_{genotype} = 13.79$, $P = 0.0002$, $F_{age} = 10.03$, $P < 0.0001$) in 3xTg DA neurons. **d** Perforated-patch recordings from spontaneously firing WT cells at baseline (black) and in the presence of apamin (1–3 nM; orange). **e** Increased firing

frequency ($n = 8$ neurons, 6 mice; two-tailed Wilcoxon, $P = 0.0078$, $W = 36.00$) and **f** $CV_{ISI}$ (two-tailed Wilcoxon, $P = 0.0078$, $W = 36.00$) in presence of apamin. **g** Depolarized apparent spike threshold in response to apamin (paired two-tailed $t$-test, $P = 0.0050$, $t = 4.031$). **h** ISI voltage trajectory averages. The effect of apamin is reminiscent of the trajectories from 3xTg neurons (panel **2k**). **i** Depolarized mAHP (paired two-tailed $t$-test, $P = 0.0022$, $t = 4.722$) and (**j**) absolute minimum ISI voltages (paired two-tailed $t$-test, $P = 0.0018$, $t = 4.853$) in apamin. Error bars represent standard error. Source data are provided as a Source Data file.

measurements indicated a strong effect of systemic SGC treatment, with significant hyperpolarization both during the injection period and up to seven days after cessation of drug (Fig. 7j), suggesting that SGC produced persistent effects on DA neuron SK channel activity and firing rate. Consistent with our immunohistochemical findings, these data suggest that SK channels remain available on the plasmalemmal surface, but SK channel-bound CaM is hyperphosphorylated in 3xTg VTA DA neurons, resulting in decreased SK channel conductance, increased spontaneous firing rate and irregularity, and hypersensitivity to somatic current injection.

## Discussion

Anecdotal evidence from AD patients suggests a possible role for dopaminergic dysfunction[88,89], yet potential druggable molecular connections between DA neuron pathophysiology and associated symptoms remain sparse[3]. To address this, we first examined ad lib fed mice for their ability to learn operant self-administration of a palatable food reward to test deficits on instrumental learning and motivation. 3xTg mice demonstrated an age-dependent impairment in learning of the task, with no significant change in consumption or motivation in the mice that did learn. We next probed single DA neurons of the VTA and found that a reduction in SK channel conductance produced irregular firing accompanied by an increase in firing rate in slices from 3xTg mice. Finally, we used a combination of whole-cell and perforated patch clamp recordings, RNA sequencing, and pharmacology to implicate CK2 hyperactivity as the likely culprit for the abnormalities.

Converging evidence has recently implicated dopaminergic dysfunction in the behavioral symptoms of AD. Findings using amyloid-only rodent models of AD indicate a DA-dependent decrease in memory formation and reward processing, concomitant with a decrease in TH+ cell bodies in the VTA and DA release in the NAc[2]. Additionally, hemizygous 3xTg mice exhibit a DA-dependent increase in basal locomotion in post- but not pre-pathological stages[5]. Although reward-motivated behavior is known to decline in dementia patients[90], it has not been a focus of investigations using AD mouse models. Here we show that non-food-restricted 3xTg mice exhibit age-dependent impairment in learning of an instrumental conditioning task for a palatable food pellet reward. Among the mice that did learn the task, there was no difference in pellets earned per session or in motivation to gain access to the reinforcer, measured by progressive ratio breakpoint. This may suggest a reduction in the reinforcing properties of the pellets in more afflicted 3xTg mice, although motivation for the reinforcer was unaffected in the mice that eventually (albeit more slowly) learned the task. Although previous models have shown conflicting data on reward learning[91,92], these studies were conducted under food and/or water restriction that would inherently confound the motivation from pure reward to incorporate a metabolic need. In total, our results are congruent with involvement of dopaminergic dysfunction in AD pathophysiology, as well as clinical data suggesting heterogeneity in disease progression and penetrance.

Our results are also consistent with the notion that DA neurons in the VTA are more vulnerable than those in the adjacent SNc in preclinical AD models[2,3], in stark contrast to the reverse association in PD[93]. While the exact mechanisms underlying preferential

pathophysiology are unknown and require further investigation, it may involve differences in the ionic mechanisms that control firing in the two nuclei[94]. One notable possibility is the influence of calbindin, a calcium-binding protein highly expressed in VTA (but not SNc) DA neurons that is thought to be protective in PD and PD models[95]. In Tg2576 mice, VTA DA neurons from prodromal mice display increased spontaneous and somatically-evoked firing[3,4] which may be due to elevated levels of calbindin. Similarly, we observed upregulated calbindin expression in 3xTg mice, which could be an adaptive mechanism to preserve survival in the face of potential pathology[3]. Our present data suggest that a decrease in SK channel activity results in hyperactive basal and stimulated DA neuron activity in 3xTg mice. SNc and VTA DA neurons and LC NE neurons differ in their SK-dependence for spontaneous firing rate and in their subtype composition[49,53,60]. Importantly, SK channel isoforms have differing sensitivity to CK2 modulation[96], but it is currently unknown how CK2 differentially tunes channels in those brain regions under physiological and maladaptive conditions. Our results from WT mice provide insights into intrinsic VTA DA neuron physiology across the lifespan. We previously reported age-dependent decreases in spontaneous and evoked firing along with a delay in rebound firing in SNc DA neurons from male WT mice beginning at around 18 months[57,97]. We also reported an apparent decrease in L-type calcium currents, but no change in (presumably SK-mediated) tail currents with age[57,97]. In contrast to the SNc, here we detected stable firing frequency (Fig. 2a, b) and gain (Fig. 2f) in the VTA with age. This indicates that the underlying ionic mechanisms may differ somewhat between the SNc and VTA in their ability to adapt to age, as tail currents are decreased in the VTA at 18 months (Fig. 3a–c) but are unaltered in the SNc. Whether there are robust age-dependent changes in VTA DA neurons in WT mice requires further investigation.

One potential explanation for the altered cell-attached firing rates in the 18-month-old 3xTg mice is survivorship bias of recorded neurons. While we do not detect reduced DA neuron numbers in the VTA of 12 mo 3xTg mice (Supplementary Fig. 7f), medial VTA DA neurons may preferentially degenerate in Tg2576 mice, as measured by a decrease in TH+ neurons[2–4]. VTA DA neurons are highly diverse, differing not only in their synaptic connectivity[98] but also intrinsic physiology[16,25,39,46,47]. It is possible the viable cells selected for recordings in aged mice were most often part of a resilient subpopulation, similar to selectively vulnerable and resilient subpopulations of SNc DA neurons in PD[70]. This could be tested in future studies in 3xTg mice using retrograde tracing, as has been done in Tg2576 mice[3]. The difference in age-associated physiological alterations between the two midbrain dopaminergic nuclei suggests that ionic mechanisms are potential factors in subpopulation-specific pathophysiology.

SK channel activation regulates both spontaneous firing and burst generation in midbrain DA neurons[17,41,99,100]. During tonic activity, SK channel activation during the ISI sets the timing for the next action potential and dampens sensitivity to synaptic input[53,58,99]. The influence of SK channels on motivated behavior may hinge on their physical proximity to NMDA receptors, where they can serve as a calcium-dependent shunt during excitatory synaptic input[58,99,100]. When a human SK channel mutation resulting in decreased channel conductance is expressed in mouse VTA DA neurons, increased NMDA

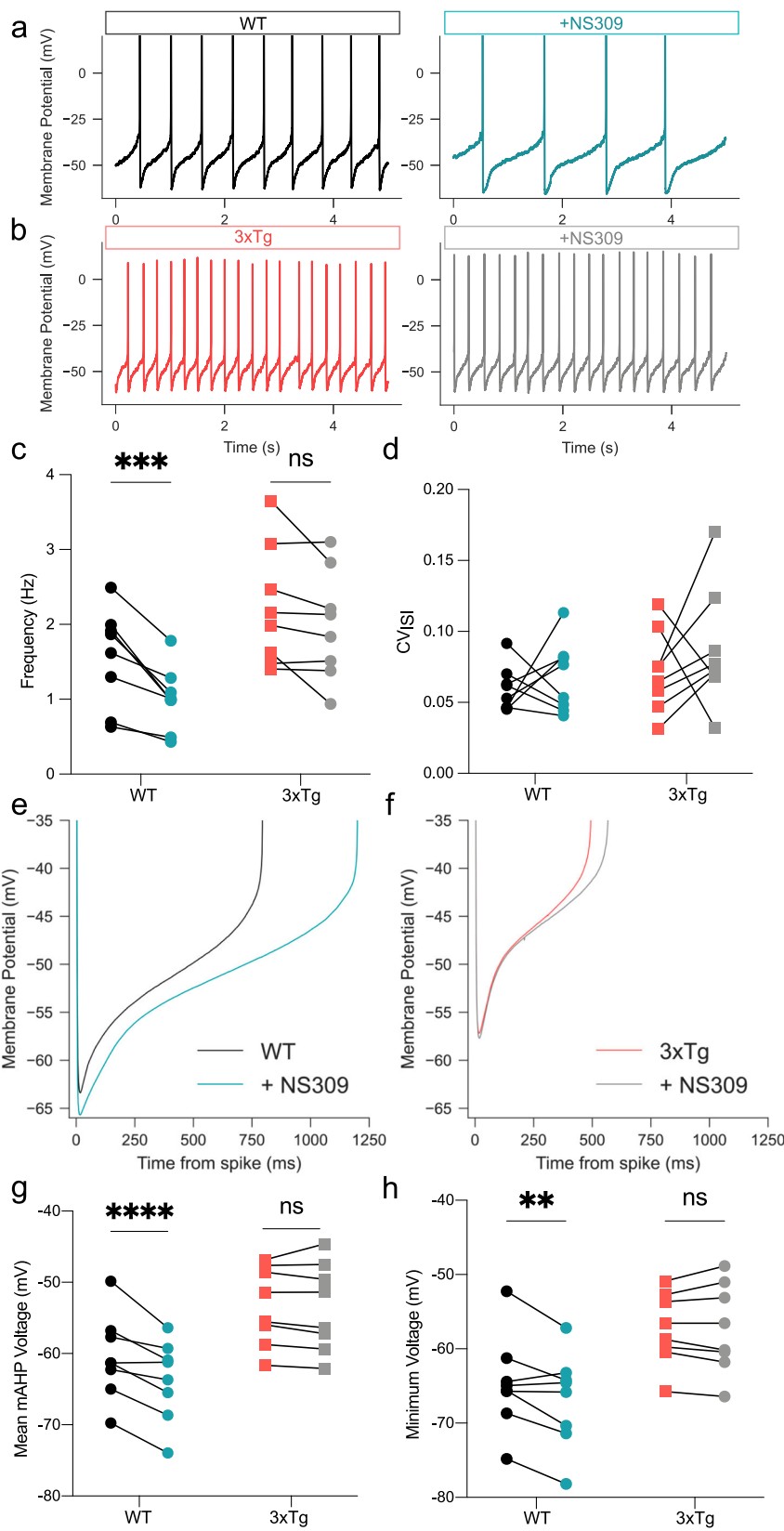

receptor-mediated currents are observed[58]. We here detected a similar result (Supplementary Fig. 6), suggesting decreased functionality of both synaptic and non-synaptic SK channels. Further studies will be necessary to fully characterize synaptic integration in 3xTg mice, as excitatory input and NMDA receptor expression are both likely altered. SK channels are thought to functionally couple to T-type calcium

channels in the SNc[55], but it is not known whether this also holds in the VTA. Roles for SK channels in non-dopaminergic cell types also include homeostatic regulation of firing contributing to behaviorally relevant tasks[101,102]. Taken together, the contributions of SK channels to DA neuron physiology position them as focal points in conditions ranging from drug addiction to neurodegenerative disease. Our current work

**Fig. 4 | NS309 efficacy is decreased in 3xTg DA neurons. a** Representative traces of WT DA neuron pacemaking in control aCSF (black) and in the presence of NS309 (1 μM) (blue). **b** Representative traces of 3xTg DA neurons in control aCSF (red) and in NS309 (gray). **c** Firing frequency was decreased by NS309 (two-way RM ANOVA, $F_{NS309} = 15.27$, $P = 0.0058$; $F_{genotype} = 5.872$, $P = 0.0459$; $F_{NS309 \times genotype} = 6.604$, $P = 0.0370$) in WT ($n = 8$ neurons, 3 mice; two-tailed Sidak's, P = 0.0007) but not significantly in 3xTg ($n = 8$ neurons, 4 mice; two-tailed Sidak's, $P = 0.0554$) neurons. **d** NS309 did not affect $CV_{ISI}$. **e** Averaged ISI voltage trajectories in WT neurons before (black) and after NS309 (blue) and (**f**) in the 3xTg group before (red) and in

the presence of NS309 (gray). **g** NS309 hyperpolarized mean medium after hyperpolarization voltages in WT (two-tailed Sidak's, $P < 0.0001$), but not 3xTg (two-tailed Sidak's, $P = 0.7795$) neurons (two-way RM ANOVA, $F_{NS309} = 9.963$, $P = 0.0160$, $F_{genotype} = 14.89$, $P = 0.0062$, $F_{NS309 \times genotype} = 43.200$, $P = 0.0003$). **h** NS309 hyperpolarized minimum ISI voltages (two-way RM ANOVA, $F_{genotype} = 11.06$, $P = 0.0127$; $F_{NS309 \times genotype} = 11.94$, $P = 0.0106$) in WT ($P = 0.0037$, two-tailed Sidak's) but not 3xTg ($P = 0.9988$, two-tailed Sidak's) neurons. Error bars represent standard error. Source data are provided as a Source Data file.

provides a plausible link between SK channel activation and DA neuron physiology in a model of AD.

Although AD is often characterized as a disorder involving Aβ-dependent synaptic failure[103,104], network hyperexcitability is also considered a hallmark[105]. Although calcium-activated potassium channels have been implicated in AD-related intrinsic and synaptic pathophysiology[27,106], the specific involvement of CK2 in synaptic alterations within AD has not been previously described. Notably, CK2 activation is thought to rely on SK channel activity[67]. This implies a potential positive feedback loop, where persistent SK channel activity caused by increased firing (or NMDA-dependent activation) could decrease the channel's activity and further potentiate hyperexcitability. Aβ has been shown to directly activate CK2 in vitro. Staining with the 6E10 antibody has been used to detect intracellular Aβ in the VTA of Tg2576 mice[4] as early as 3 months of age. However, 6E10 also detects soluble amyloid precursor protein (sAPP)[107], and thus these results may indicate increased soluble species specifically in the VTA. sAPP has been implicated in presynaptic function and intrinsic neuronal excitability[7,107–110]. The resulting circuit dysfunction may drive adaptive intrinsic deficits, akin to proposals for SNc DA neurons in PD[94]. Furthermore, SK-bound CK2 may serve as a potential link between intrinsic and synaptic imbalances in AD, given its proposed role in tuning SK channel calcium sensitivity and thus synaptic integration and homeostatic firing rates. A disparity between firing rate homeostasis and synaptic integration in AD has been proposed as the key mechanism driving the shift from prodromal to clinically apparent stages of AD[108], and CK2 inhibition abolishes spike adaptation in some neurons[77], suggesting a target for this form of pathophysiological adaptation.

SK channels are voltage-independent potassium channels that open in response to a local rise in intracellular calcium. In response to $Ca^{2+}$ binding, CaM undergoes a conformational change that promotes the formation of a complex between CaM and a binding domain on the SK channel[111]. This in turn stabilizes an intrinsically disordered fragment (R396-M412) to open the pore and produce $K^+$ efflux[111,112]. This conformational change also depends on PIP2 binding[66], which is the process promoted by the positive allosteric modulator NS309[66]. Critically, this enhancement is diminished by CaM phosphorylation at threonine 79 (T79), the target of CK2. Our Patch-seq and immunofluorescence experiments identify upregulated CK2 expression in single DA neurons and suggest its involvement in SK channel dysregulation, as previously reported in other brain regions[77,79] and in vitro[63]. Further, upregulated CK2 apparently alters firing parameters in neurons from 3xTg mice, which we later confirmed using pharmacology. The decreased effect of NS309 (Fig. 4) suggested a potential hyperphosphorylation of SK channel-bound CaM T79 in DA neurons from 3xTg mice, which we then detected at the protein level (Fig. 6d, e).

Direct pharmacological inhibition of CK2 had a dramatic effect in 3xTg DA neurons but did not alter physiological parameters in WT cells (Fig. 7). This suggests that VTA DA neurons maintain minimal basal levels of CK2-dependent phosphorylation that then switches to a hyperphosphorylated state in 3xTg mice. While the exact mechanism triggering increased CK2 activity requires further investigation, CK2

activity is known to be stimulated by Aβ[84], which accumulates intracellularly in neurons from 3xTg mice[28,113]. Furthermore, CK2 upregulation precedes hyperphosphorylation of tau in AD patients, and increased CK2 expression is highly correlated with hyperphosphorylated tau[81] as it inhibits SET which subsequently promotes tau phosphorylation[83]. Thus, an interaction between Aβ and tau may further promote this mechanism, both in 3xTg mice and humans with AD. We propose that CK2-depedent phosphorylation of SK-bound CaM triggers physiological adaptations in VTA DA neurons of mice that express aberrant levels of Aβ and phosphorylated tau (Fig. 8).

CK2 is a pleiotropic kinase. $CK2\alpha^{-/-}$ mice do not develop proper hearts or neural tubes, and mortality occurs mid-gestation[114]. CK2 is also strongly associated with AD[80,83], PD[115], and various forms of cancer. The pharmacological profile of CK2 is well described, and clinical trials investigating its antagonist CX4945 are ongoing for basal cell carcinoma, in healthy subjects for tolerability, and for medulloblastoma (ClinicalTrials.gov: NCT03897036; NCT05817708; NCT03904862, respectively). Our data suggest that CK2 is a potential mediator of DA neuron hyperactivity in AD. Pharmacological inhibition of CK2 with SGC rectified irregular firing patterns in 3xTg DA neurons, while application of SGC to WT cells did not result in observable changes in spontaneous firing rate or rhythmicity. This suggests that basal phosphorylation levels may be low in VTA DA neurons, leaving them primed for physiological (or pathological) adaptation. As dopaminergic hyperactivity is a hallmark of multiple conditions including addiction and psychiatric disorders, these findings point toward a potential role of CK2 in modulating DA neuron firing across diverse contexts.

## Methods
### Animals
Male and female WT and 3xTg-AD mice were bred in-house at the Oklahoma Medical Research Foundation in accordance with the Institutional Animal Care and Use Policies. Mice were housed on a 12 h light/12 h dark reverse light cycle (lights off at 0900). Humidity was maintained between 40% and 60%, and temperature was held between 18 and 23 °C. All mice had *ad libitum* access to food and water, except for those undergoing attention gating. Founders for 3xTg (MMRRC #034830) and WT controls were obtained from Dr. Salvatore Oddo[28]. 3xTg mice were homozygous for all three humanized transgenes ($APP_{Swe}$, $PSEN1_{M146V}$, and $Tau_{P301L}$). Due to the hybrid background (C57BL/6;129×1/SvJ;129S1/Sv embryo injected in a B6;129), a separate line on the same hybrid background, not carrying the transgenes, was maintained in parallel and used as control (WT). Breeder genotyping was outsourced (Transnetyx). All experiments and assays included male and female mice, which were roughly balanced in each experiment. As there were no indications of any sex effects, data were pooled.

### Operant conditioning behavior
Food and water were available ad libitum in the home cage at all times for the mice that underwent behavioral testing, unless otherwise noted (see attention gating methods). Operant testing was conducted in 3-h daily sessions beginning two hours into the dark cycle in operant chambers that were housed in sound-dampening cabinets equipped

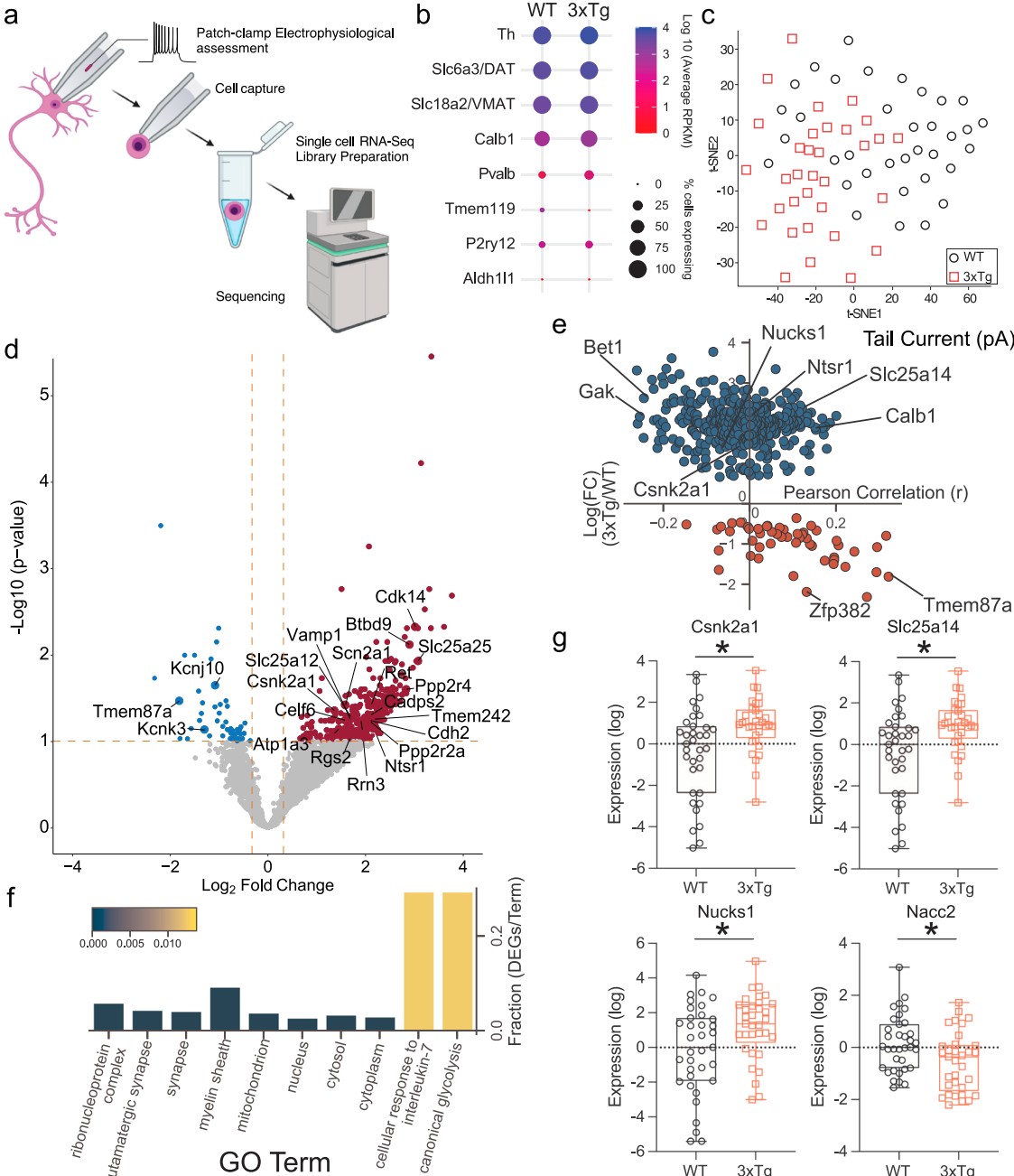

**Fig. 5 | Patch-seq analysis indicates upregulated *Csnk2a1* transcription. a** For Patch-seq, recording from an individual neuron was followed by cell extraction with the recording pipette. Cell contents were captured, and each individual cell RNA-Seq library was prepared and sequenced. **b** Individual cells were assessed for markers of DA neurons and potential contaminating cells. **c** t-SNE projection of each cell using all expressed genes demonstrates some genotype-dependent segregation. **d** Differentially expressed genes (DEGs) were determined by *t*-test with Benjamini Hochberg Multiple testing correction and a fold change filter of >|1.25|. **e** Plotting of differential expression fold change (FC) versus Pearson correlation to tail current amplitude indicated genes correlated and anticorrelated with tail current amplitude. *Csnk2a1* (−0.03172 Pearson r) and the gene coding for casein kinase 2 dependent protein *Nucks1* (−0.0687, Pearson r) were anticorrelated with tail current amplitude and upregulated in 3xTg DA neurons. **f** Gene ontology pathway analysis from DEGs indicates glycolysis and cytokine response as hits. **g**. Examples of individual DEGs, including *Csnk2a1* (0.0443, Benjamini−Hochberg multiple testing correction), *Slc25a14* (0.0475, Benjamini−Hochberg multiple testing correction), *Nucks1* (0.0993, Benjamini-Hochberg multiple testing correction), and *Nacc2* (0.0873, Benjamini−Hochberg multiple testing correction), *false discovery rate <0.1. Box and whisker plots define the inner quartile range and whiskers extend to minimum and maximum values, center line corresponds to median. Source data are provided as a Source Data file. Created in BioRender. Freeman, W. (2023) BioRender.com/k29t209.

with fans to minimize external noise. Modular operant chambers (Lafayette Instruments) had two nose-poke holes with a food receptacle between them and a sonalert speaker across from the food receptacle. For each chamber, one nose poke hole was designated as the correct hole and was illuminated by a stimulus light within it during the session. Responses in the correct hole resulted in reinforcer delivery when the response requirement was met. Responses in the nose poke hole designated as incorrect had no consequence. The correct side was balanced across chambers to decrease side-preference bias. Training began at fixed ratio 5 (FR5); five nose-pokes

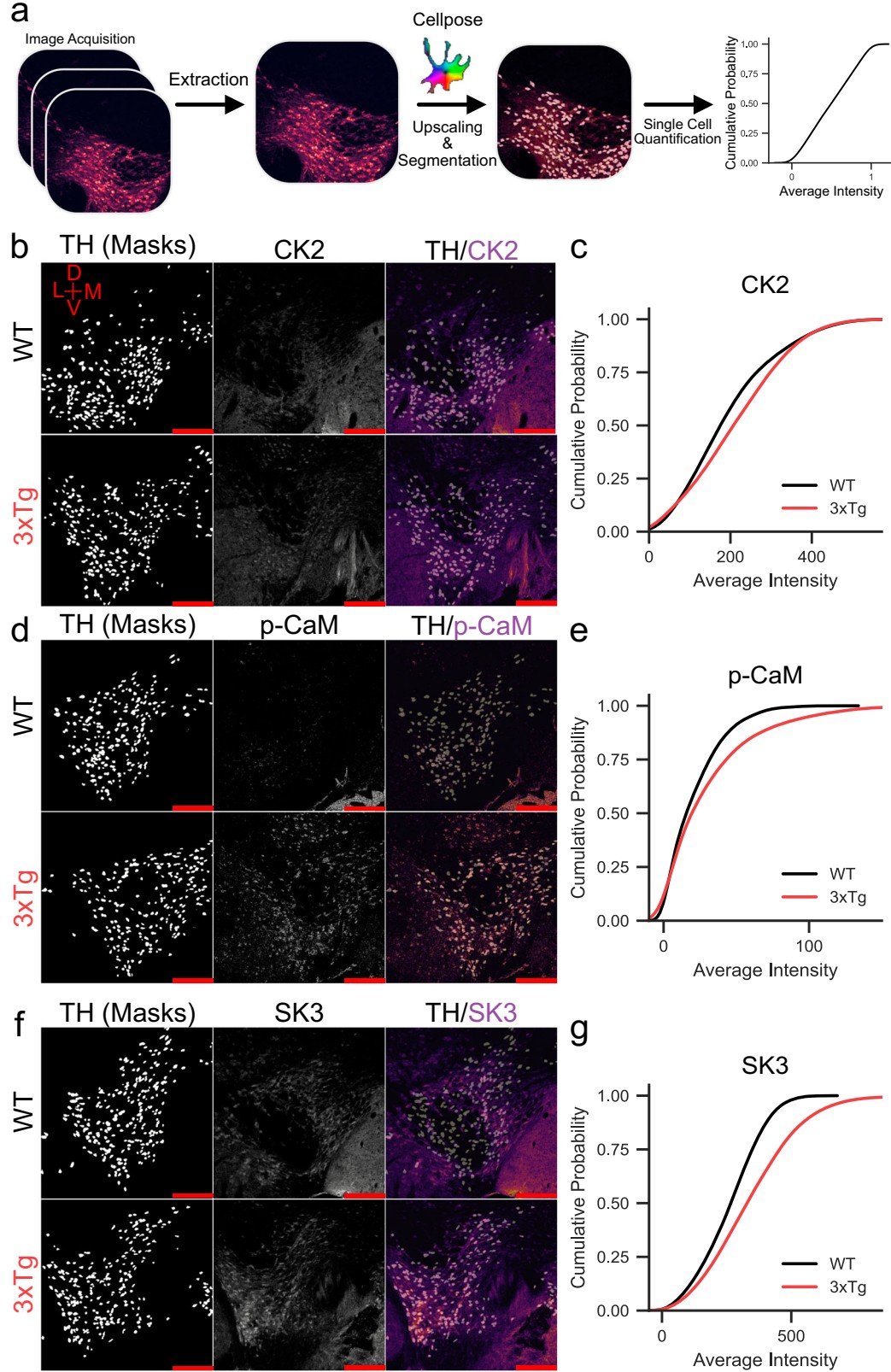

in the correct hole resulted in administration of a 20-mg palatable food pellet (chocolate-flavored chow pellet, Bio-Serve), activation of a cue-light in the receptacle, and a 5-s tone. There was a 30-s timeout after pellet delivery, during which time no responding was reinforced and the stimulus light in the correct side nose poke hole was extinguished. The response requirement was increased across sessions (FR5, FR10, FR20, FR30) when accuracy for the correct side nose poke hole was at

least 70% on any single day (accuracy defined as correct side nose pokes/total nose pokes on both sides × 100), with at least 1 day at each response requirement. Acquisition was defined as the first day of training that a mouse obtained 70% accuracy for the correct side nose poke hole. If a mouse did not reach acquisition criterion by the end of the fourteenth session, the mouse was removed from the study and deemed a "non-learner" (Fig. 1b). After meeting criterion at FR30 and

**Fig. 6 | CK2, p-CaM, and SK3 protein expression are increased in 3xTg VTA DA neurons. a** Workflow for cell segmentation analysis following immunohistochemistry. Confocal z-stack images were acquired, then the brightest slice of the channel of interest was chosen for analysis. DA neurons were segmented on TH staining from the same plane using the Cellpose 'cyto v3' model following upscaling (see Methods). Intensities within each segmented cell were calculated, quantified as average intensity across single cells, and plotted as cumulative probabilities. **b** Staining for CK2 indicated a slight rightward shift in the cumulative probability curve (**c**) of casein kinase 2 (CK2) intensity in tyrosine hydroxylase (TH) positive cells from 3xTg mice (Kolmogorov−Smirnov test, $D = 0.1183$, $p < 0.0001$, $n = 2353$ WT and 2187 3xTg cells, 6 mice). **d** Staining for phosphorylated calmodulin (p-CaM) indicated a moderate rightward shift in the cumulative probability curve (**e** Kolmogorov−Smirnov test, $D = 0.1202$, $p < 0.0001$, $n = 2612$ WT and 2193 3xTg neurons, 6 mice). **f** SK channel upregulation in 3xTg mice was indicated by a strong rightward shift in the cumulative probability of small conductance calcium-activated potassium channel 3 (SK3) intensity in single cells (**g** Kolmogorov−Smirnov test, $D = 0.2284$, $p < 0.0001$, $n = 2914$ WT and 3165 3xTg neurons, 6 mice). Red scale bars = 200 μm. Source data are provided as a Source Data file.

maintaining stable behavior, mice were subjected to a progressive ratio (PR) session, which was terminated after 60 min without earning a reinforcer or 5 h total. During this PR test session, the FR requirement increased with each pellet earned according to the following schedule: 1, 2, 4, 6, 9, 12, 15, 20, 25, 32, 40, 50, 62, 77, 95, 118, 145, 178, 219, 268, 328, 402, 492, and 603. Breakpoint was defined as the last successfully completed response requirement before the session was ended.

## Locomotion

Open field Opto-Varimex chambers (44.5 cm × 2.9 cm × 3.2 cm, Columbus Instruments) were used to collect locomotor data on 3-month WT ($n = 9$) and 3xTg ($n = 9$) mice and on 12-month WT ($n = 12$) and 3xTg ($n = 13$) mice. Mice were habituated to the behavioral room for 30 min and were tested in the open field for 1 h. The center of the chamber was defined as the interior 39% of the arena area (the middle 10 out of 16 infrared beams on each axis). Data analysis was performed on the entire 60-min session.

## Attention gating

Male and female WT ($n = 9$) and 3xTg ($n = 6$) mice (11–15 months old) were used in Pavlovian conditioning and attentional gating assays with methods adapted from previously published work[58]. Mice were pre-exposed to 20-mg palatable chocolate pellets in their home cage 3 days prior to the start of Pavlovian conditioning. For each day of training, food was removed from the home cage at 8:30 a.m. (30 min before lights out). Mice were habituated to the testing room 5-h after food was removed (1:30 p.m.) for 30-min prior to the session. After the daily session (approximately 57 min each, days 1–7), mice were returned to their home cage with food and water ad libitum until 8:30 a.m. the next day. Each mouse performed one session per day for 8 consecutive days in sound attenuating operant conditioning chambers (Lafayette Instruments) as outlined for operant pellet self-administration. On days 1–7, each session consisted of 50 Pavlovian conditioning trials total, 25 CS^Low and 25 CS^High, where CS^Low was defined as 10 s of an auditory tone at one frequency followed by pellet delivery in 12% of the trials and CS^High defined as 10 s of an auditory tone at a different frequency with pellet delivery in 100% of the trials. Each trial occurred on a variable inter-trial interval (ITI), with trials presented on average every 60 s (range 40–80 s) in random orders for ITI and CS condition for each session. Tone frequencies were counterbalanced across conditions (CS^High and CS^Low). On day 8, mice were tested in an attentional gating assay of 30 trials total, 15 CS^High and 15 CS^Novel, where CS^High maintained the above criteria and CS^Novel was defined as CS^High criteria concurrent with 10 s of a flashing house light positioned inside of the chamber on the opposite side of the food receptacle. Every trial on day 8 (CS^High and CS^Novel) resulted in pellet delivery.

Dependent variables for days 1–7 included latency for head entry into the food receptacle, calculated for both CS^High and CS^Low. Day 8 analysis included a comparison of attention index, calculated as the difference in latency for head entry into the receptacle between CS^Novel and CS^High as an average of 3-trial bins for each condition (CS^Novel - CS^High). Day 8 data from four WT mice were removed due to lack of pellet retrieval (in both conditions) that precluded analysis.

## Brain slice preparation

For each electrophysiological experiment, the mouse was anesthetized with isoflurane and decapitated. The brain was removed rapidly and sliced in an ice-cold cutting solution containing the following (in mM): 110 choline chloride, 2.5 KCl, 1.25 $Na_2PO_4$, 0.5 $CaCl_2$, 10 $MgSO_4$, 25 glucose, 11.6 Na-ascorbate, 3.1 Na-pyruvate, 26 $NaHCO_3$, 12 N-acetyl-L-cysteine, and 2 kynurenic acid. Horizontal (200 μm) slices containing the ventral midbrain at the level of the medial terminal nucleus of the accessory optic tract ($mt$) were collected and immediately transferred to a holding chamber containing artificial cerebrospinal fluid (aCSF) containing (in mM) 126 NaCl, 2.5 KCl, 1.2 $MgCl_2$, 2.4 $CaCl_2$, 1.2 $NaH_2PO_4$, 21.4 $NaHCO_3$, and 11.1 glucose. Holding chamber aCSF was supplemented with (in mM) 1 Na ascorbate, 1 Na pyruvate, 6 N-acetyl-L-cysteine, and 0.01 MK-801, to minimize excitotoxicity associated with slicing. Slices recovered for thirty minutes at 32 °C followed by an additional 30 min or more at room temperature prior to recording. For the SGC-preincubation experiments, 1 μM was added to the holding chamber, and slices were maintained in this solution for 1–3 h prior to recording. Neurons were recorded up to ten hours after slicing.

## Electrophysiology

Slices were transferred to a recording chamber where they were perfused with warmed aCSF (Warner Instruments inline heater, 32–34 °C) at a rate of 2 mL/min via gravity or a peristaltic pump (Warner Instruments). Slices were visualized under Dödt gradient contrast (DGC) optics on an upright microscope (Nikon). Putative VTA dopaminergic neurons were identified first by location (>100 μm medial to $mt$ at the level of the fasciculus retroflexus [$fr$]). A subset of recorded neurons were labeled with biocytin and imaged on a confocal microscope (Zeiss LSM 710) or imaged under DGC on the rig for localization (Supplementary Fig. 11). Electrophysiological criteria for VTA DA neuron identification included slow, spontaneous firing in the cell-attached configuration (0.5–8 Hz), wide (>1.1 ms) spikes[116], inactivation time constants of the A-type potassium current between 30 and 300 ms[48], and a long (>250 ms) rebound delay after a hyperpolarizing step (−100 pA) in current clamp[46,47]. SNc DA neurons were recorded <50 μm from $mt$ (Supplementary Fig. 2a), and electrophysiological identification parameters included slow (1–5 Hz) spontaneous firing and wide (>1.1 ms) spikes in the cell attached configuration, fast-inactivating A-type potassium conductance, and hyperpolarization-activated inward current ($I_H$) >100 pA. Locus coeruleus NE neurons were identified by large cell bodies (>20 μm) directly medial to the mesencephalic trigeminal nucleus, rostral to the border of the fourth ventricle and with electrophysiological parameters including slow (0.25–6 Hz) spontaneous firing, slow-inactivating A-type potassium conductance, and membrane resistance <100 MΩ (−77 $V_{hold}$). For cell-attached and whole-cell recordings, patch pipettes were pulled from thin-wall glass (World Precision Instruments or Warner Instruments) and had resistances of 2.5–3 MΩ when filled with an intracellular solution containing (in mM): 135 K gluconate, 10 HEPES, 5 KCl, 5 $MgCl_2$, 0.1 EGTA, 0.075 $CaCl_2$, 2 ATP, 0.4 GTP, pH 7.35[117]. Whole-cell internal solution had a liquid junction

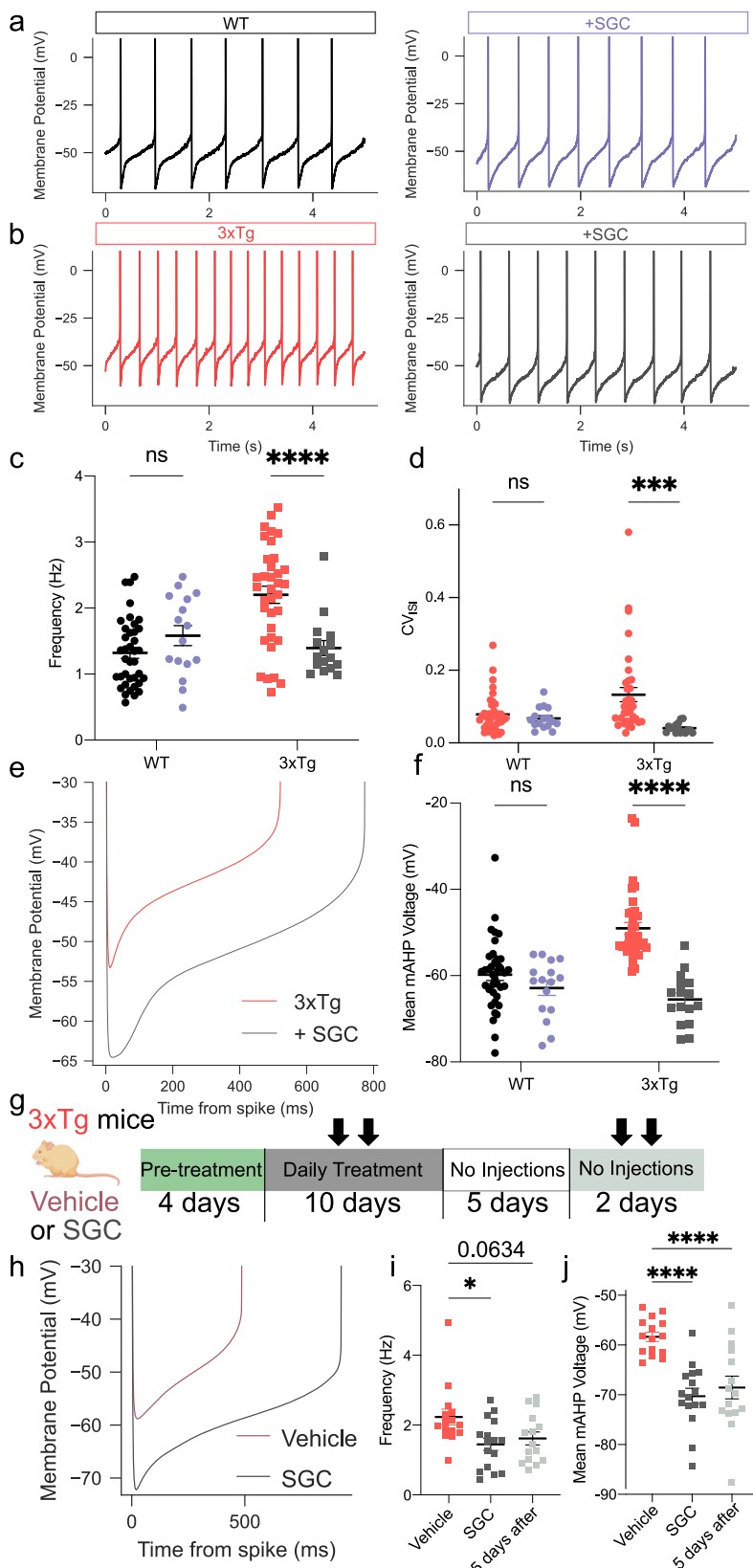

potential of −17.1 mV, which was calculated with the stationary Nernst–Planck equation using LPJcalc (https://swharden.com/LJPcalc) and corrected offline. Perforated-patch pipettes were pulled from standard wall glass (Warner Instruments) and had resistances of 3.5–4.5 MΩ. The tip was first filled with (in mM) 140.5 KCl, 7.5 NaCl, 10 HEPES (pH 7.0) and then backfilled with the

same solution containing 1–2 μg/mL gramicidin-D. Recordings generally began when perforation was adequate to display action potential overshoot of 0 mV, and recordings were terminated if break-in occurred. Patch-seq pipettes were pulled from thin-walled glass similar to whole-cell recording pipettes except with a slightly smaller tip (3–4 MΩ) to allow efficient nucleated patch formation[69].

**Fig. 7 | Casein kinase 2 inhibition restores firing properties in 3xTg DA neurons. a** Representative firing trace from a naïve WT neuron (black) and one pre-incubated with SGC (1 µM, purple). **b** Representative trace from a naive 3xTg neuron (red) and one pre-incubated with SGC (gray). **c** SGC decreased firing frequency in 3xTg ($n = 35$ neurons, 8 naïve mice and $n = 16$ neurons, 3 SGC treated mice; two-tailed Sidak's, $P < 0.0001$) but not WT ($n = 38$ neurons, 10 naive mice and $n = 16$ neurons, 3 SGC treated mice; two-tailed Sidak's, $P = 0.1140$) neurons; $F_{genotype} = 5.988, P = 0.0161; F_{SGC} = 3.514, P = 0.0637; F_{genotype \times SGC} = 18.93, P < 0.0001$. **d** SGC decreased $CV_{ISI}$ in 3xTg (two-tailed Sidak's, $P = 0.0002$, same mice as in **c**) but not WT (two-tailed Sidak's, $P = 0.9796$) DA neurons ($F_{SGC} = 9.502, P = 0.0026; F_{genotype \times SGC} = 8.034, P = 0.0055$). **e** Averaged inter-spike interval (ISI) voltage trajectories of naive 3xTg neurons and ones pre-incubated with SGC. **f** SGC hyperpolarized the mAHP in 3xTg ($P < 0.0001$) but not WT ($P = 0.3225$) neurons (two-way ANOVA, $F_{genotype} = 6.349, P = 0.0133; F_{SGC} = 36.74, P < 0.0001; F_{genotype \times SGC} = 17.22, P < 0.0001$, same mice as in **e**). **g** Schematic of in vivo treatment paradigm. 3xTg mice received either vehicle or SGC (1 mg/kg, i.p.) twice a day

for four days (pre-treatment), daily treatment for ten days, followed by cessation of treatment for up to seven days. Electrophysiology experiments took place either during the daily treatment phase or final no-drug phase (denoted with arrows) to test short-term and persistent effects of SGC treatment. **h** Averaged ISI voltage trajectories of vehicle-treated (burgundy) and SGC-treated (gray) 3xTg neurons. **i** Systemic SGC treatment decreased firing frequency (one-way ANOVA, $F = 4.374$, $P = 0.0187$) in 3xTg mice ($n = 15$ neurons, 2 vehicle mice) during the daily injection phase ($n = 16$ neurons, 2 mice; two-tailed Dunnett's, $P = 0.0134$) and trended toward significance up to seven days following injections ($n = 15$ neurons, 2 mice; two-tailed Dunnett's, $P = 0.0634$). **j** SGC treatment hyperpolarized the mean medium after hyperpolarization (one-way ANOVA, $F = 14.53, P < 0.0001$, same mice and cells as in **i**) during the daily injection phase (two-tailed Dunnett's, $P < 0.0001$), an effect that persisted after injections were terminated (two-tailed Dunnett's, $P = 0.0002$). Error bars represent standard error. Source data are provided as a Source Data file. Created in BioRender. Beckstead, M. (2023) BioRender.com/y52l770.

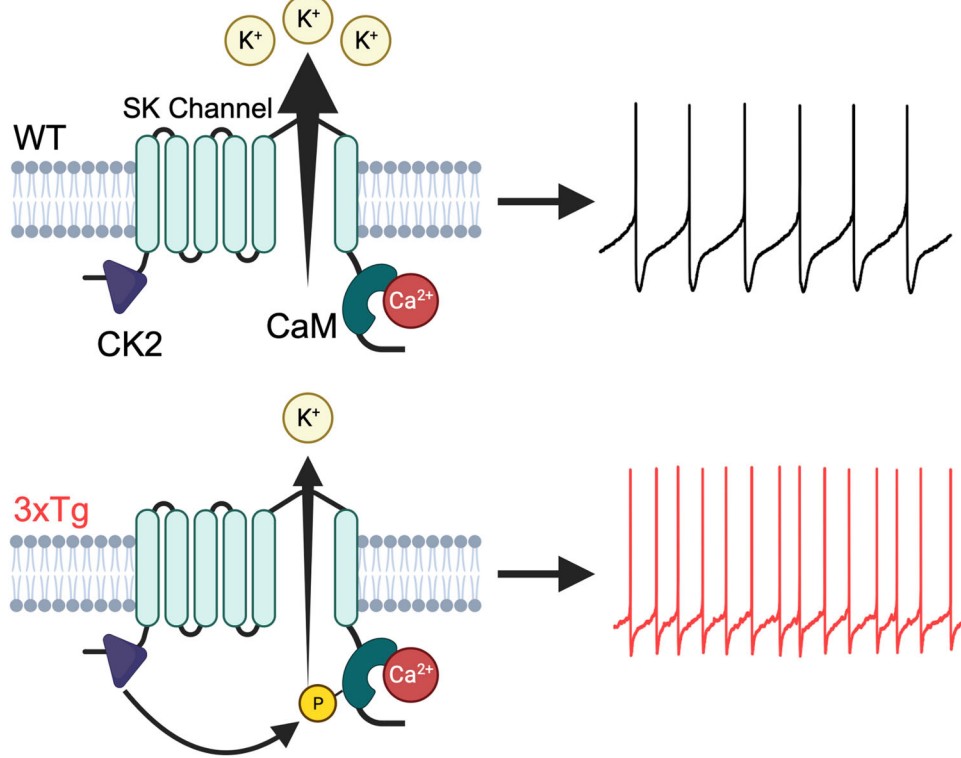

**Fig. 8 | Proposed mechanism of hyperexcitability in 3xTg VTA DA neurons.** Normally, VTA DA neurons maintain a balance between CK2 and PP2A resulting in low basal phosphorylation of CaM. In 3xTg mice, hyperactive CK2 results in

hyperphosphorylated CaM, effectively decreasing SK channel calcium binding affinity and channel opening. This results in increased firing rate and irregularity in 3xTg mice. Created in BioRender. Beckstead, M. (2023) BioRender.com/j31r759.

For evoked NMDA EPSCs, MK-801 was not included in the recovery chamber, and a 0 $Mg^{2+}$ aCSF was used (126 NaCl, 2.5 KCl, 3.6 $CaCl_2$, 1.2 $NaH_2PO_4$, 21.4 $NaHCO_3$ and 11.1 D-glucose), plus DNQX (10 µM), picrotoxin (100 µM), hexmethamonium (100 µM), CGP56999a (100 nM), and sulpiride (50 nM). Recording pipettes were filled with a QX-314-containing K-gluconate based internal (140.5 K-gluconate, 7.5 NaCl, 10 HEPES, 0.1 EGTA, 2 MgATP, 0.21 NaGTP, 5 Br-QX-314). Cells were voltage clamped at −72 mV (corrected). A bipolar stimulating electrode was placed at least 200 µm caudal to the recorded cell. EPSCs were evoked every 6 s. A stable baseline of five minutes was acquired prior to bath application of apamin (10 nM). Charge was calculated as AUC using the Numpy trapezoid function. NMDA currents were confirmed by sensitivity to MK-801 (10 µM). All pipettes were pulled on a vertical PC-10 or PC-100 puller (Narishige International).

### Chemicals and pharmacological agents

NS309 was purchased from Alomone Labs. Apamin was purchased from Echelon Biosciences. SGC-CK-2 was purchased from Tocris. All other chemicals were purchased from Sigma-Aldrich.

### Electrophysiological data acquisition and analysis

Recordings were acquired with a MultiClamp 700B amplifier and digitized with an Instrutech ITC-18 board. AxoGraph (v1.7.6 and v1.8.0) and LabChart (ADInstruments) software were used for recording. Voltage-clamp recordings were low-pass filtered online at 6 kHz and digitized at 20 kHz. Current clamp recordings were low-pass filtered at 10 kHz and digitized at 20 kHz. Experimenters were not blinded to the group during data collection; however, all data was batch analyzed and then sorted based on genotype. All analyzes were conducted offline via custom scripts, or implementation of eFEL in Fig. 2e, f, in Python (3.9).

Code used for analyzes is publicly available at the following site with example files and Jupyter notebooks (https://github.com/heblanke/Electrophysiology_analysis_CK2_study). Cell-attached recordings were filtered using a Savitzky-Golay filter and action currents were detected in the negative direction. In current clamp, spikes were generally detected at the upward crossing of −10 mV. ISI times and voltages were collected by subtracting the first spike time from the times between each pair of spikes and were resampled at 1000 points spanning each ISI to allow for averaging across ISIs and across different cells. The minimum voltage value was taken as the minimum of the average ISI for a cell. The average voltage across the mAHP was calculated as the average voltage from 10 ms after the minimum point to 100 ms after the minimum point of the ISI. The ISI voltage trajectories illustrated are grand averages across all cells in a group. Tail current AUC was calculated using the trapezoidal rule implemented in NumPy. A-current inactivation time constants were determined by fitting a bi-exponential curve to currents using the SciPy curve fit function. Apparent threshold values were determined by averaging spike voltages for a given cell and determining the voltage at which 10 mV/ms was crossed. Action potentials were binned into 10 ms windows and averaged across bins to measure frequency and CV of ISI in order to avoid confounds of rate drift. Representative traces of action potentials were truncated at either +10 mV (Figs. 2, 3, and 6) or at +20 mV (Fig. 4).

## Patch-seq

Patch-seq required slight modifications to recording parameters to minimize confounds in both electrophysiological recordings and sequencing. Patch-seq internal was a modification of whole-cell internal solution containing (in mM) 124 K gluconate, 10 HEPES, 5 KCl, 5 $MgCl_2$, 0.1 EGTA, 0.075 $CaCl_2$, 20 μg/mL glycogen, and 0.5 units/μL RNase inhibitor (M0314L, New England Biolabs). The voltage step protocol used to measure the tail current was unchanged. To minimize recording time before cell extraction, additional electrophysiological parameters were not recorded. After recording was complete, light negative pressure was applied to the pipette until the nucleus was visible at the tip of the pipette, or for approximately three minutes. Next, the recording pipette was slowly retracted along the Z-axis. Successful extractions included an obvious nucleated patch[69]. After the nucleus was visualized above the slice, the pipette was rapidly retracted from the bath, and the cellular contents within the pipette were deposited in a 0.2 mL PCR tube containing 0.5 μL of NEB Next lysis buffer and 0.25 μL NEB murine RNase inhibitor. Tube contents were then rapidly spun down and flash frozen on dry ice. Samples were stored at −80 °C until RNASeq libraries were generated.

## RNASeq library generation

Directional libraries were constructed using NEBNext Single Cell/Low Input RNA Library Prep Kits from Illumina following manufacturer instructions. In brief, poly-adenylated mRNA was magnetically captured and then eluted from the oligo-dT beads. mRNA was then fragmented and reverse-transcribed, and the cDNA was purified using SPRISelect beads (#B23318, Beckman Coulter). Purified cDNA was eluted in 50 μL 0.1 × TE buffer. Adaptors were diluted and ligated to cDNA, and the cDNA was amplified by 28 cycles of PCR. SPRISelect beads were used to select for proper library size. Purified library quality was then assessed on a tapestation (#4200, Agilent Technologies) using HSD1000 screentapes (#5067−5584; Agilent Technologies). Due to the high number of amplification cycles, occasionally libraries required additional rounds of SPRISelect purification. Qubit 1 × dsDNA HS Assay kit (#Q33230, Thermo Fisher Scientific) was used to quantify libraries. Libraries were pooled and sequenced 2 × 100 bases on an Illumina NovaSeq X. RNA sequencing data generated in this study are available through the NCBI Gene Expression Omnibus using accession number GSE273040.

## RNASeq analysis

Reads were trimmed and aligned before differential expression statistics and correlation analyzes in Strand NGS software package (v4.0; Strand Life Sciences). Samples with reads were aligned against the full mm10 transcriptome RefSeq build (2013.04.01). Alignment and filtering criteria included the following: adapter trimming, fixed 5-bp trim from 5′ and 3′ ends, a maximum number of one novel splice allowed per read, a minimum of 90% identity with the reference sequence, a maximum 5% gap, and trimming of 3′ ends with $Q < 30$. All duplicate reads were then removed. Normalization was performed with the DESeq2 algorithm. Transcripts with an average read count value ≥1 in at least 80% of the cells in at least one group were considered expressed at a level sufficient for quantitation. Cells positive DA neuron markers *TH*, *Slc6a3*, and *Slc18a2* and negative for markers of other cell types *Pvalb*, *Tmem119*, *P2ry12*, *Aldh1l1* were retained, 68 cells in total. For statistical analysis of differential expression, a *t*-test with Benjamini–Hochberg multiple testing correction (BHMTC) was performed with a False Discovery Rate cutoff of <0.1. For those transcripts meeting this statistical criterion, an absolute fold change >1.25 cutoff was used to eliminate those genes that were statistically significant but unlikely to be biologically significant and orthogonally confirmable because of their very small magnitude of change. t-SNE were performed in Strand NGS (version 4.0)[118]. To compare electrophysical parameters to changes in gene expression, Pearson correlations were computed using the scipy.stats pearsonr function in Python. Gene ontology analysis was performed using GOATOOLS[119]. QC data for the Patch-seq is provided in Supplementary Fig. 9.

## Immunohistochemistry

Mice were deeply anesthetized with 2% 2,2,2-tribromoethanol in injectable saline via IP injection, followed by transcardial perfusion with PBS, then 4% paraformaldehyde (PFA). Mice were decapitated and brains were extracted and post-fixed in 4% PFA for 16 h and cryoprotected in 30% sucrose in PBS for three days, or until brains had completely sunk. Brains were then embedded in optimal cutting temperature (Tissue-Tek) and frozen on dry-ice. Striatal and midbrain sections were cut coronally on a sliding cryotome (CryoStar) at 40 μm and slices washed in PBS. All mice used for immunohistochemistry were harvested and brain tissue processed in parallel.

For all histology, autofluorescence was quenched with 5 mg/mL $NaBH_4$ in water, washed with PBS, and subsequently slices were permeabilized in 0.2% PBST (Triton X-100). Tissue was blocked in 5% bovine serum albumin and 0.5% normal donkey serum. Primary antibodies were diluted as follows: chicken anti-tyrosine hydroxylase (TH, 1:750, Abcam), rabbit anti-Kcnn3 (SK3, 1:200, Alomone, Thermo-Fischer), rabbit anti-casein kinase 2 beta 1 (CK2, 1:200, Invitrogen), rabbit anti-phospo-calmodulin (p-CaM, 1:100, Invitrogen), and rat anti-DAT (1:750, Millipore), and incubated for 72 h at 4 °C. Secondary antibodies were diluted as follows: anti-chicken AlexaFluor 488 (1:750, JacksonImmuno), anti-rabbit AlexaFluor 594 Plus (1:500, Invitrogen), and anti-rat AlexaFluor 649 (1:500, JacksonImmuno) incubated at 4 °C overnight. Each well for processing only contained two primary antibodies, anti-TH and a second antibody from the list above. Secondary-only controls were used to confirm no aberrant fluorescence. Striatal slices were always stained with anti-TH and anti-DAT. Slices were mounted on Diamond White Glass charged slides (Globe Scientific), dried, rinsed with water, and air dried again prior to application of mounting medium (Prolong Diamond) and coverslips.

## Confocal microscopy and image analysis

Images were acquired at 10× on a Zeiss 880 confocal microscope. Within groups, images were acquired using identical settings. All image analyzes included at least three sections per mouse and three mice per group. Images were exported as .czi files, cropped to contain only the VTA, and imported into Python using the aicsimageio CziReader

implementation to convert the file to a Numpy array. For segmentation analysis, arrays were sliced for the brightest image in the non-TH channel, which was extracted for both the TH and non-TH channels. The anti-TH channel was used for DA neuron segmentation using CellPose3[120,121]. Images were first upsampled using the 'denoise' function (via 'upsample_cyto3') and then DA neurons were segmented using the 'cyto3' model. To calculate unmanipulated fluorescence intensity, the cell masks were then resized using the 'skimage.transform resize' function. The correct size mask was then used to calculate single cell intensity values, by calculating the pixel intensities within cell masks for CK2, p-CaM, and SK3. Average intensity for single cells were computed and cumulative probability curves generated from the data using the Seaborn 'kdeplot' function. For determination of total striatal and midbrain TH and DAT intensity, sum intensity projections were calculated, normalized to the max value for each image and total intensity from the frame was reported. Image quantification was conducted on raw images. For display purposes in representative images, if a channel intensity was increased it was done so uniformly across genotypes.

## Statistical analysis

Statistics on electrophysiological data were analyzed in Prism (versions 9 and 10), and statistics and calculations for RNAseq were conducted in Python, Strand NGS, and Prism. All electrophysiology parameters collected were batch analyzed to minimize bias, except for wash-in pharmacological manipulations. Normality was assessed using the Shapiro−Wilk test. Unpaired and paired two-tailed *t*-tests were used for normal two-group comparisons. Mann−Whitney and Wilcoxon tests were used for non-normal unpaired and paired two-group comparisons, respectively. For cumulative probability graphs, two group Kolmogorov−Smirnov tests were used. Two-way ANOVAs with Sidak's post-hoc test for multiple comparisons or one-way ANOVAs followed by Dunnett's test were conducted where indicated. For graphs where an omnibus test was conducted, asterisks representing significant group effects (i.e., genotype) were placed adjacent to the legend. All error bars represent standard error of the mean (s.e.m.) unless otherwise noted.

## Reporting summary

Further information on research design is available in the Nature Portfolio Reporting Summary linked to this article.

## Data availability

The RNA sequencing data generated in this study have been deposited in the NCBI Gene Expression Omnibus database under accession code GSE273040. The imaging and electrophysiology data are too large to deposit alongside analysis code; however, raw data are available through contacting the corresponding author. The data generated in this study to reproduce figures are provided in the Supplementary Information/Source Data file. Source data are provided with this paper.

## Code availability

Custom scripts to analyze data can be located at https://github.com/heblanke/CK2_SK_ephys_and_image_analysis[122].

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

## Acknowledgements

This work was supported by grants from the NIH (R21 AG072811 and R01 AG052606 to MJB, F31 AG079620 to HEB, P30AG050911 and R01AG059430 to WMF), the Department of Veterans Affairs (I01 BX005396 and IK6BX006033), the Presbyterian Health Foundation, and the Oklahoma Center for Adult Stem Cell Research, a program of TSET. We would like to thank Dr. Ezequiel Marron for insightful questions about the project, and Drs. Stephanie Gantz, Christopher Ford, and Matthew Higgs for their invaluable comments and suggestions. We are immensely grateful to Dr. Sarah Ocañas for initial support troubleshooting the Patch-seq procedure and the OMRF Clinical Genomics Core for assistance sequencing. We would like to also thank Dr. Holly Van Remmen for sharing research materials and the OMRF Imaging Core, especially Julie Crane and Ben Fowler. Finally, we thank Matt Croxall at Lafayette Instruments for assistance in analysis of the attention gating experiment. Schematics in Figs. 1, 5, 7, 8, and Supplementary Fig. 5 were generated in part with BioRender.com.

## Author contributions

H.E.B. conducted electrophysiology experiments, collected samples for Patch-seq experiments, prepared and imaged immunofluorescence samples, and wrote custom analysis scripts. H.E.B., N.T.C., and K.D.P. prepared RNA sequencing libraries. K.A.C., A.N.M., and M.I.T. conducted behavioral assays. H.E.B. and M.J.B. wrote the manuscript. H.E.B., M.J.B., A.L.S., and W.M.F. designed studies and analyzed data. All authors approved final version of manuscript.

## Competing interests

The authors declare no competing interests.
