## [Transparent Peer Review file · Nature Communications]

VTA dopamine neurons are hyperexcitable in 3xTg-AD mice due to casein kinase 2-dependent SK channel dysfunction

Corresponding Author: Dr Michael Beckstead

Version 0:

Reviewer comments:

Reviewer #1

(Remarks to the Author)

In this paper by Blankenship et al., the authors characterised the functional deficits in VTA dopamine neurons in the 3xTg mouse model. In particular, they investigated alterations in reward-related operant learning, changes in the excitability of VTA DA neurons and alterations in the SK channel conductance in these cells. Overall, they observed that in 3xTg mice the DA neurons are hyper-excitable, and this is related to a reduction in the SK channel and loss of AP-afterhyperpolarization. With Patch-seq sequencing directly from DA cells, they detected alterations in the SK channel modulator casein kinase 2 (CK2). With this in mind, they propose an interesting theory in which the enhanced activity of CK2 in DA neurons of 3xTg mice results in phosphorylation of Cam and reduced SK-channel conductance. Although this theory has not been investigated thoroughly, treatments with a CK2-inhibitor can restore SK channels and the excitability of 3xTg neurons. Overall, this is an interesting, high-quality work that adds to the known DAergic deficits in AD models by a mechanism to explain functional alterations and DA-relevant behavioural deficits. For these reasons it merits consideration by the journal, provided that the points arisen below will be addressed satisfactorily.

Major comments

- The authors link the behavioral deficits (Fig.1) with dysfunction in DA signalling in areas such as the NAc. Yet, info in this mouse model about the onset of DA neuron deficits is clearly missing. It would be of particular relevance to identify the age of onset of DA deficits and repeat the behavioral test before this age. This would clearly prove the role of the VTA and exclude the role of Abeta or tau that, I assume, are enhanced since birth. Please also describe at which stage of the disease the ages analysed correspond to (3, 6, 12 and 18 months).
- Line 227-228: 'not been previously reported': Please correct this as it is inaccurate, SK-mediated tail currents have been recorded from Tg2576 VTA dopamine neurons in La Barbera et al., 2021, where the authors also show reduction in tail current amplitude (but not current density).
- Line 297: the conclusion that SK channels might be less sensitive to NS309 is premature. Could these results be due to reduced expression of SK channels, rather than reduced functioning due to altered phosphorylation? are there indications of the levels of SK channels in these neurons (i.e. with confocal microscopy?). Alternatively, to test for reduced sensitivity the authors could try a higher concentration of NS309 in 3xTg mice. in the same line of thinking, why are the data (in Line 335-336) not shown? Proving that the channel expression levels are not reduced can indirectly confirm the author's theory that the reduced current measurement is due to reduced function of these channels.
- Line 302-305: this is an interesting theory to explain the results but there are no experiments to support it. Are there any specific targets downstream of CK2 to investigate for increased levels of phosphorylated forms, such as CaM itself, with confocal microscopy?
- What are the effects of SGC in vivo on the behavioral experiment? is this able to restore motivation in these mice?
- Line 417: the results of hypersensitivity to current injection suggest that at minimal stimulation by afferent glutamatergic

inputs the DA neurons of 3xTg would respond more with burst firing (due to lack of SK current?) and therefore release more DA in the projection targets, including the NAc. One would assume a higher, and not reduced, reward learning in these animals. Yet the mice behave differently. How do the authors explain these results? What is the VTA status in these mice? are the DA neurons degenerated like in the Tg2576 mouse? Are there evidence of reduced DA levels in the NAc, despite the hyperexcitability of DA neurons, to explain the behavioral results of the study?

- Line 552-555: Please explain further the link between CK2 and Abeta levels or tau. Additionally, the authors should try to explain the selective deficits of SK channels (and therefore CK2) in the VTA, in contrast to SNpc and LC neurons that are unchanged. What might be the basis of this selective dysfunction of SK channels in the VTA? Is it possible that this area accumulates more Abeta or tau? Is there any difference in these toxic species between VTA and SNpc DA neurons in 3xTg mice? Are SK channels expressed differently in VTA compared to the other areas?.

- The manuscript would also be highly improved if authors showed that at an age in which DA neurons in the VTA function normally the levels of CK2 are also normal between WT and Tg.

- A table of all electrophysiological data for the various ages and stats would be highly appreciated. This is because the representation of statistics in the figures are a bit unclear. For example in Fig.2b I assume that the 2 stars in WT vs 3xTg indicate the results of the ANOVA for genotype ($P=0.0025$), and the 3 stars over 12 months show the results of post-hoc tests? Please clarify.

Also, for example in Figure 2B, $N=576$ cells; figure 2F, $N=222$ cells. Please specify for each genotype and age the number of cells analysed. In all the legends, specify the number of animals used.

- The manuscript requires a thorough revision of the literature:

- Line 32. Add La Barbera et al., *Prog Neurobiol* 2021:202:102031. This is a paper that recently described DA neuron hyperexcitability in Tg2576, in line with what the authors describe in 3xTg and definitely merits acknowledgement

- Line 35, add refs of the 'sparse' works

- Line 41, ref 15: please add original works, particularly from the Surmeier lab studying the involvement of Ca-induced toxicity in SNpc DA neurons

- Line 43: the distinct mechanism of firing between DA neurons in VTA and SNpc are well described by this work: Khaliq & Bean (*J Neurosci.* 201026;30(21):7401-13)

- Line 68, Ref22: citing this work is not sufficient, these authors study the effect of receptor antagonists systemically, not locally on the NAc. Please add other relevant works

- Line 405: pre-clinical models and clinical trials: please add refs or clinical trial number

Minor comments

- Lines 29-30: this sentence seems misplaced: motor signs suggest deficits in the nigrostriatal dopaminergic system, whereas the previous and following sentences focus more on deficits associated with the mesocorticolimbic system from the VTA

- Line 35: 'cortical and hippocampal studies': this is unclear, please explain

- Line 63-64: rephrase: i don't see how the expression of tau or abeta 'permits exorption into neuropsychological abnormalities'.

- Line 288: I recommend changing this title, from the results it appears that 3xTg DA neurons are unaltered by NS309.

- Line 310: Patch-seq technique: are these confirmed DA neurons? since the electrophysiology was minimal in patch-seq experiments, how can the authors tell that these are DA neurons? Were these cells tested for TH or DAT expression?

- Line 499-500: please avoid phrases such as 'first data' or 'for the first time'.

- Line 570: Male vs female: results from males and females are pooled together? did the authors check for differences between sexes?

- Please cite figure 5F.

- In all the figures: please uniform graph dimension and font. in figure 6g, please increase the graphical description of SGC treatment.

- Uniform DA and dopamine.

Reviewer #2

(Remarks to the Author)

In this study by Blankenship et al., the authors investigated the role of VTA dopamine neurons dysfunction in a mouse model of Alzheimer's disease. They identified the dysregulation of the calcium-activated small conductance potassium channel, SK3 as a contributor to altered physiology of VTA dopamine neurons in a 3XTg mouse model of AD. They further demonstrated that elevated casein kinase 2 signaling is a major contributor to SK channel dysregulation in this model system. Overall, the study is elegant in its design and provides a comprehensive analysis of the transcriptional alterations in dopamine neurons in the context of an animal model of AD. Although my enthusiasm for this manuscript and its novel findings are high, there are phenotypic and mechanistic considerations that should be addressed to provide a better understanding between the relationship of altered dopamine neuron physiology and behavioral dysfunction related to this AD model.

Specific comments:

1. The authors use a simple instrumental task as a behavioral model to implicate dopamine neuron dysfunction in the 3xTg mouse model. Reduced learning and motivation in an operant task are typically associated with a reduction in VTA

dopamine neuron activity, yet the authors identify a decrease in tail currents and increased excitability of dopamine neurons. This would suggest an increase in dopamine that should enhance motivational processes. It should be noted however, that hyperactivation of the dopamine system is also associated with altered attentional processing and sensory gating. Because of the cognitive impairments associated with the 3xTg mouse model, which is likely the result of cortical and hippocampal dysfunction, this experiment does not provide a logical link between the behavior and the observed electrophysiological changes observed. Because elevated dopamine is linked to psychosis-related phenotypes and psychosis is prevalent in AD, behavioral analyses should be designed with the consideration of these symptom domains. Indeed, reducing SK-mediated tail current in VTA dopamine neurons is associated with alterations in sensory-motor gating, attentional disruption, and increased sensitivity to the psychomimetic actions of NMDA receptor antagonists (<https://doi.org/10.1016/j.neuron.2013.07.044>). Therefore, to make the study more relevant to neuropsychiatric symptoms associated with AD, as indicated in the first line of the abstract, the authors should perform at least one behavioral experiment that has relevance to the proposed roles of dopamine in behavioral domains relevant to psychosis.

2. As stated in the discussion, SK channels couple to NMDA-type glutamate receptors in the hippocampus (10.1038/nn1449) forming a calcium feedback loop. This relationship was also demonstrated in VTA dopamine neurons (<https://doi.org/10.1016/j.neuron.2013.07.044>). Casein Kinase 2 activity enhances NMDA receptor function (doi: 10.1038/5680) and regulates NMDA receptor subunit composition (DOI 10.1016/j.neuron.2010.08.011 and <http://dx.doi.org/10.1016/j.celrep.2013.02.011>). This could result in a scenario in which reduced SK currents would lead to enhanced NMDA receptor currents that may already be further enhanced because of elevated CK2 signaling in the 3xTg model resulting in an additive effect of potentiated NMDA receptors and reduced SK-mediated feedback. Because of this, it would benefit the study greatly if the authors would address whether NMDA-evoked currents are significantly potentiated in the 3xTg model.

Reviewer #3

(Remarks to the Author)
Comments are in the attachment.

Version 1:

Reviewer comments:

Reviewer #1

(Remarks to the Author)
The Authors seriously considered my previous comments so that the Ms is now improved.

Reviewer #2

(Remarks to the Author)
The authors have addressed my previous concerns and the manuscript is greatly improved. The novel findings of this study make an important contribution to the field.

Reviewer #3

(Remarks to the Author)
Thank you for conducting the additional locomotor studies. The results convincingly demonstrate that 3xTg mice have increased rather than decreased locomotion. I have two comments to add:
1). When comparing the residency plots between WT and 3xTg, the center regions of WT mice seem more uniform (with values close to zero). This suggests that 3xTg mice tend to avoid the center, which is particularly evident in the "dark" region in Fig 1e compared to 1d. This observation contrasts with the results of Figure 1i and the authors' conclusion that "3xTg mice also spent significantly more time in the center of the locomotor area."
2). The tornado plots seem to show that WT mice's activity decreased over time, likely due to exploration and habituation with the chamber. However, 3xTg mice's activity appears uniform over time. Quantifying the dispersion over time could serve as a behavioral metric to differentiate the phenotype.
Overall, these results align with the cognitive impairment (there is no reward associated here) observed in 3xTg mice.

Thank you for the organization of the code in your GitHub repo and for providing better annotation and examples. Since all the functions are contained in the "funcs_for_ck2_sk_study.py" file, it is not necessary to redefine functions in the notebooks (instead, import only those needed from the .py file). This would help reduce errors due to replication and allow readers to focus more on the analysis pipeline. If figures 2e and 2f are using the eFEL pipeline instead of a custom script, mentioning that in the corresponding method section would be appropriate.

The quality and quantity of patch-seq data, as well as the associated analyses, have been greatly improved. A few comments:

1). Lines 234 to 236: "4,165 genes passed the criteria (read count >1 in >80% of samples from at least one group) for expression and transcriptome profiles, and WT and 3xTg cells generally separated by t-distributed stochastic neighbor embedding (t-SNE; Figure 5c)." I suggest splitting this sentence into two. The second half, "WT and 3xTg cells generally separated by t-distributed stochastic neighbor embedding (t-SNE; Figure 5c)," is poorly written and not accurate. The two

groups are not generally (linearly) separable in the t-SNE map. However, based on your DEG analysis, a generic classifier in the original (4165) dimension would give you good classification.

2). Please add a formal description of how you are calculating "Euclidean correlation" in the method section. If it is standard Euclidean distance, it should not be between -1 and 1, regardless of normalization. Please provide justification for why it is preferred over standard Pearson correlation (or demonstrate that the latter provides comparable results). Please replace "Euclidean similarity" (in Fig 5d caption and supp. Table 1) with "Euclidean distance" if they refer to the same metric.

3). There is a recent large single-nucleus sequencing dataset of midbrain DA neurons in mice publicly available (Salmani et al. (2024), "Transcriptomic atlas of midbrain dopamine neurons uncovers differential vulnerability in a Parkinsonism lesion model," eLife 12). Integrating an analysis between your data and this dataset could provide further insights. For example, you could check if VTA region-specific Calb1 (Figure 3B in Salmani 2024) is expressed in most patch-seq data, or if patch-seq VTA data are uniformly distributed among the reference single-nucleus data (by "mapping" the former onto the latter) to check for sampling bias. In a future study, you might consider using a similar approach to compare VTA DA neurons in 3xTg mice and a control group using high-throughput single-nucleus sequencing, as VTA-specific midbrain DA neurons are readily separable from the rest. This could be particularly relevant when comparing different age groups, assuming the regional differences hold across ages.

Version 2:

Reviewer comments:

Reviewer #3

(Remarks to the Author)

Authors have addressed my concerns from previous round. Clarity of this manuscript has been improved. No more comment.

We thank the reviewers for their careful reading of our manuscript. New experiments have been added to address comments and suggestions, including a new and larger cohort of Patch-seq cells (Fig. 5), new immunohistochemistry experiments (Fig. 6 and Supp. Fig. 7), and behavioral assays (Fig. 1, Supp. Figs. 1 and 5), along with updated interpretation and addition of citations. These additions have strengthened the manuscript and further support our finding that CK2 dysregulates SK channel activity in ventral tegmental area dopamine neurons in the 3xTg mouse model of Alzheimer's disease. Response to specific comments follows.

Reviewer #1:

In this paper by Blankenship et al., the authors characterised the functional deficits in VTA dopamine neurons in the 3xTg mouse model. In particular, they investigated alterations in reward-related operant learning, changes in the excitability of VTA DA neurons and alterations in the SK channel conductance in these cells. Overall, they observed that in 3xTg mice the DA neurons are hyper-excitabile, and this is related to a reduction in the SK channel and loss of AP-afterhyperpolarization. With Patch-seq sequencing directly from DA cells, they detected alterations in the SK channel modulator casein kinase 2 (CK2). With this in mind, they propose an interesting theory in which the enhanced activity of CK2 in DA neurons of 3xTg mice results in phosphorylation of Cam and reduced SK-channel conductance. Although this theory has not been investigated thoroughly, treatments with a CK2-inhibitor can restore SK channels and the excitability of 3xTg neurons. Overall, this is an interesting, high-quality work that adds to the known DAergic deficits in AD models by a mechanism to explain functional alterations and DA-relevant behavioural deficits. For these reasons it merits consideration by the journal, provided that the points arisen below will be addressed satisfactorily.

Major comments:

The authors link the behavioral deficits (Fig.1) with dysfunction in DA signalling in areas such as the NAc. Yet, info in this mouse model about the onset of DA neuron deficits is clearly missing. It would be of particular relevance to identify the age of onset of DA deficits and repeat the behavioral test before this age. This would clearly prove the role of the VTA and exclude the role of Abeta or tau that, I assume, are enhanced since birth. Please also describe at which stage of the disease the ages analysed correspond to (3, 6, 12 and 18 months).

To address this comment, we have completed and include in Fig. 1 operant food self-administration experiments at 3 months of age, which is prior to development of electrophysiological deficits. Consistent with this, 3xTg mice display normal operant self-administration at this age, acquiring the task as rapidly as WT controls. We also now report locomotor activity for WT and 3xTg mice at 3 and 12 months, which shows increased distance traveled and time in center of an open field in the 3xTg mice. We have also expanded our description of the established pathological progression in 3xTg mice (3 months, prepathological; 6 months, detectable amyloid-plaques; 12 months, intracellular hyperphosphorylated tau aggregates; 18 months, neuronal death in previously assayed brain regions; Lines 63-67).

Line 227-228: 'not been previously reported': Please correct this as it is inaccurate, SK-mediated tail currents have been recorded from Tg2576 VTA dopamine neurons in La Barbera et al., 2021, where the authors also show reduction in tail current amplitude (but not current density).

We apologize for the oversight and now include this finding and citation. We additionally have clarified that our goal here was to use SK specific pharmacology to confirm the channel identity of the tail current in adult mice.

Line 297: the conclusion that SK channels might be less sensitive to NS309 is premature. Could these results be due to reduced expression of SK channels, rather than reduced functioning due to altered phosphorylation? are there indications of the levels of SK channels in these neurons (i.e. with confocal microscopy?). Alternatively, to test for reduced sensitivity the authors could try a higher concentration of NS309 in 3xTg mice. in the same line of thinking, why are the data (in Line 335-336) not shown? Proving that the channel expression levels are not reduced can indirectly confirm the author's theory that the reduced current measurement is due to reduced function of these channels.

We have now added an extensive immunofluorescence study (Fig. 6). The results indicate that phosphorylated calmodulin (p-CaM) levels are significantly higher in tyrosine hydroxylase (TH) positive cells in 3xTg VTA compared to WT. There is a greater density of SK channels in 3xTg DA versus controls. These results

complement our NS309 experiment results and suggest that SK channels are available in both WT and 3xTg DA neurons, but phosphorylation of calmodulin decreases their function in 3xTg mice.

Line 302-305: this is an interesting theory to explain the results but there are no experiments to support it. Are there any specific targets downstream of CK2 to investigate for increased levels of phosphorylated forms, such as CaM itself, with confocal microscopy?

New confocal microscopy results indicate that phospho-calmodulin is significantly elevated in 3xTg DA neurons compared to WT (Fig. 6).

What are the effects of SGC in vivo on the behavioral experiment? is this able to restore motivation in these mice?

We did not do this experiment but we don't believe so. Please see next response.

Line 417: the results of hypersensitivity to current injection suggest that at minimal stimulation by afferent glutamatergic inputs the DA neurons of 3xTg would respond more with burst firing (due to lack of SK current?) and therefore release more DA in the projection targets, including the NAc. One would assume a higher, and not reduced, reward learning in these animals. Yet the mice behave differently. How do the authors explain these results? What is the VTA status in these mice? are the DA neurons degenerated like in the Tg2576 mouse? Are there evidence of reduced DA levels in the NAc, despite the hyperexcitability of DA neurons, to explain the behavioral results of the study?

We completely agree with this assessment and have now included microscopy results suggesting that, similar to Tg2576 mice, 3xTg DA axons are degenerating in the ventral striatum (Supplemental Figure 7a-c). Additionally, we include results from the midbrain that total TH intensity is decreased (Supplemental Figure 7d-e). In contrast to Nobili et al 2017, we did not detect a change in overall DA neuron count (Supplemental Figure 7f), however we did not perform an extensive stereological study here. Concerning SGC, the in vivo effects would likely be complicated by the suspected denervation. We now propose that hyperactivity of VTA DA neurons through phosphorylation of SK bound calmodulin may be a homeostatic mechanism to increase dopamine output in the face of neurite die-back, at which point SGC administration in vivo may have a paradoxical negative effect.

Line 552-555: Please explain further the link between CK2 and Abeta levels or tau. Additionally, the authors should try to explain the selective deficits of SK channels (and therefore CK2) in the VTA, in contrast to SNpc and LC neurons that are unchanged. What might be the basis of this selective dysfunction of SK channels in the VTA? Is it possible that this area accumulates more Abeta or tau? Is there any difference in these toxic species between VTA and SNpc DA neurons in 3xTg mice? Are SK channels expressed differently in VTA compared to the other areas?

We agree that the regional selectivity defies a simple explanation, though we now suggest (Lines 330-334) it could depend on differential expression of CK2 and SK channels (Wolfart et al 2001, PMID: 11331374) and the ability of CK2 to tune SK channel activity across brain regions dependent upon SK channel subtypes (Nam et al. 2021, PMID: 33422768). The connection between CK2 upregulation and the pathology produced in the 3xTg-AD model is an ongoing area of research in our lab, and we believe it could be driven by upregulated excitatory synaptic drive, a phenomenon previously reported in many models of AD, and a possibility we now discuss (Lines 357-361). We have also added discussion regarding the interaction between CK2 and tau (Lines 403-406). While it is possible that aBeta has a direct interaction with CK2, as previously reported and discussed (Lines 401-402), we now propose a runaway feedback loop that begins with synaptic dysfunction. We now include a new experiment showing that NMDA EPSCs are less sensitive to apamin in 3xTg mice than in WT (Supp. Fig. 6), which indicates that both synaptic and intrinsic mechanisms are altered through SK channel dysfunction. We are currently conducting a separate study to fully characterize excitatory and inhibitory synapses in the 3xTg model. Additionally, we have added clarifying context to Lines 374-375 citing La Barbera et al 2021 who detected intracellular aBeta in VTA dopamine neurons in the Tg2576 mice by 3 months of age.

The manuscript would also be highly improved if authors showed that at an age in which DA neurons in the VTA function normally the levels of CK2 are also normal between WT and Tg.

Regrettably we did not test this, as we believed the other immunohistochemistry experiments were a higher priority.

A table of all electrophysiological data for the various ages and stats would be highly appreciated. This is because the representation of statistics in the figures are a bit unclear. For example in Fig.2b I assume that the 2 stars in WT vs 3xTg indicate the results of the ANOVA for genotype (P=0.0025), and the 3 stars over 12 months show the results of post-hoc tests? Please clarify.

We have now clarified the statistics within each figure caption and in the Methods. We have also added two statistics tables at the end of the manuscript, one with age-dependent electrophysiological changes (Supp. Table 2) and a second that focuses on the firing effects of different conditions in 12-month-old mice (Supp. Table 3).

Also, for example in Figure 2B, N=576 cells; figure 2F, N=222 cells. Please specify for each genotype and age the number of cells analysed. In all the legends, specify the number of animals used.

We now specify cells and mice used in the main and supplemental figure captions.

The manuscript requires a thorough revision of the literature: Line 32. Add La Barbera et al., Prog Neurobiol 2021;202:102031. This is a paper that recently described DA neuron hyperexcitability in Tg2576, in line with what the authors describe in 3xTg and definitely merits acknowledgement

We apologize for the oversight. La Barbera et al 2021 is now cited here and throughout the paper in relevant locations.

Line 35, add refs of the 'sparse' works

We have added a selection of papers where the authors investigate single neuron dysfunction late into pathological disease stages of AD models.

Line 41, ref 15: please add original works, particularly from the Surmeier lab studying the involvement of Ca-induced toxicity in SNpc DA neuron

Clinical and pre-clinical citations have been added.

Line 43: the distinct mechanism of firing between DA neurons in VTA and SNpc are well described by this work: Khaliq & Bean (J Neurosci. 201026;30(21):7401-13)

We now cite this review here.

Line 68, Ref22: citing this work is not sufficient, these authors study the effect of receptor antagonists sistemically, not locally on the NAc. Please add other relevant works

We now cite studies showing that responding for palatable food reward is reduced by nucleus accumbens receptor blockade and lesions.

Line 405: pre-clinical models and clinical trials: please add refs or clinical trial number

In vivo pre-clinical studies have been cited and clinical trial numbers listed for ongoing CK2 inhibition studies.

Minor Comments:

Lines 29-30: this sentence seems misplaced: motor signs suggest deficits in the nigrostriatal dopaminergic system, whereas the previous and following sentences focus more on deficits associated with the mesocorticolimbic system from the VTA

We agree, this sentence has been removed.

Line 35: 'cortical and hippocampal studies': this is unclear, please explain

We have reworded this section to clarify studies of pathophysiology in hippocampus and cortex.

Line 63-64: rephrase: i don't see how the expression of tau or abeta 'permits exporation into neuropsychological abnormalities'.

This transition has been rephrased.

Line 288: I recommend changing this title, from the results it appears that 3xTg DA neurons are unaltered by NS309.

We apologize for the oversight. The title of this section has been renamed to summarize a decrease in NS309 sensitivity in the 3xTg VTA DA neurons.

Line 310: Patch-seq technique: are these confirmed DA neurons? since the electrophysiology was minimal in patch-seq experiments, how can the authors tell that these are DA neurons? Were these cells tested for TH or DAT expression?

Figure 5 now includes a dotplot indicating that the Patch-seq neurons that were analyzed highly express dopaminergic markers TH, DAT, and VMAT, and do not express Pvalb (GABAergic), Tmem119 and Pr2y12 (Microglial), or Aldh111 (astrocytic) markers.

Line 499-500: please avoid phrases such as 'first data' or for the first time'.

These phrases have been removed.

Line 570: Male vs female: results from males and females are pooled together? did the authors check for differences between sexes?

We apologize and have now provided more clarity in the Methods. All experiments include male and female mice. Initial work was focused on detecting alterations between male and female 3xTg mice. However, after determining that there were no differences in the critical assays (behavior, spontaneous firing frequency, and coefficient of variation of the interspike interval) we pooled all male and female measurements for all experiments.

Please cite figure 5F.

It has now been cited (Line 244).

In all the figures: please uniform graph dimension and font. in figure 6g, please increase the graphical description of SGC treatment.

We have now tried to standardize all graph dimensions, fonts, and line thicknesses. We have also increased the size of the graphical description of the SGC treatment.

Uniform DA and dopamine.

It is now DA in all cases, except for use of the word "dopaminergic."

Reviewer #2:

In this study by Blankenship et al., the authors investigated the role of VTA dopamine neurons dysfunction in a mouse model of Alzheimer's disease. They identified the dysregulation of the calcium-activated small conductance potassium channel, SK3 as a contributor to altered physiology of VTA dopamine neurons in a 3XTg mouse model of AD. They further demonstrated that elevated casein kinase 2 signaling is a major contributor to SK channel dysregulation in this model system. Overall, the study is elegant in its design and provides a comprehensive analysis of the transcriptional alterations in dopamine neurons in the context of an animal model of AD. Although my enthusiasm for this manuscript and its novel findings are high, there are phenotypic and mechanistic considerations that should be addressed to provide a better understanding between the relationship of altered dopamine neuron physiology and behavioral dysfunction related to this AD model.

Specific comments:

1. The authors use a simple instrumental task as a behavioral model to implicate dopamine neuron dysfunction in the 3xTg mouse model. Reduced learning and motivation in an operant task are typically associated with a reduction in VTA

dopamine neuron activity, yet the authors identify a decrease in tail currents and increased excitability of dopamine neurons. This would suggest an increase in dopamine that should enhance motivational processes. It should be noted however, that hyperactivation of the dopamine system is also associated with altered attentional processing and sensory gating. Because of the cognitive impairments associated with the 3xTg mouse model, which is likely the result of cortical and hippocampal dysfunction, this experiment does not provide a logical link between the behavior and the observed electrophysiological changes observed. Because elevated dopamine is linked to psychosis-related phenotypes and psychosis is prevalent in AD, behavioral analyses should be designed with the consideration of these symptom domains. Indeed, reducing SK-mediated tail current in VTA dopamine neurons is associated with alterations in sensory-motor gating, attentional disruption, and increased sensitivity to the psychomimetic actions of NMDA receptor antagonists (<https://doi.org/10.1016/j.neuron.2013.07.044>). Therefore, to make the study more relevant to neuropsychiatric symptoms associated with AD, as indicated in the first line of the abstract, the authors should perform at least one behavioral experiment that has relevance to the proposed roles of dopamine in behavioral domains relevant to psychosis.

To address this concern we conducted a sensory gating experiment and show the results in Supp. Fig. 5. Both groups equally learned the behavior task, and although a slight trend was evident we did not detect a change in sensory gating between WT and 3xTg mice. We have therefore modified the claims in the introduction. Additionally, we have now added locomotor data to Fig. 1 which indicates that, consistent with literature, 3xTg mice locomote more than their WT counterparts and spend more time in the center of an open field. This rules out a potential deficit in movement or exploratory behavior contributing to apparently decreased operant learning deficits in 3xTg mice.

2. As stated in the discussion, SK channels couple to NMDA-type glutamate receptors in the hippocampus (10.1038/nn1449) forming a calcium feedback loop. This relationship was also demonstrated in VTA dopamine neurons (<https://doi.org/10.1016/j.neuron.2013.07.044>). Casein Kinase 2 activity enhances NMDA receptor function (doi: 10.1038/5680) and regulates NMDA receptor subunit composition (DOI 10.1016/j.neuron.2010.08.011 and <http://dx.doi.org/10.1016/j.celrep.2013.02.011>). This could result in a scenario in which reduced SK currents would lead to enhanced NMDA receptor currents that may already be further enhanced because of elevated CK2 signaling in the 3xTg model resulting in an additive effect of potentiated NMDA receptors and reduced SK-mediated feedback. Because of this, it would benefit the study greatly if the authors would address whether NMDA-evoked currents are significantly potentiated in the 3xTg model.

We greatly appreciate this suggestion. We have now included an experiment where we block SK channels while evoking NMDA receptor-mediated EPSCs. In the WT group we saw an increase in NMDA charge transfer, similar though not as large as in Soden et al 2016, but in the 3xTg group, this increase was nonexistent. These results suggest that both NMDA-associated SK channels and the SK channels that govern pacemaking are dysfunctional in 3xTg DA neurons. We did not conduct an NMDA wash-on experiment as many factors could contribute to current amplitude, however our next study will be a full characterization of excitatory and inhibitory synaptic transmission in 3xTg mice.

Reviewer #3:

This manuscript aims to identify pathological mechanisms in single VTA DA neurons of AD mice model (3xTg) at two ages, using patch-clamp recording, single-cell sequencing and behavior test. Results show DA neurons in disease model have hyperexcitability associated with decrease of SK channel currents which in turn was affected by upregulation of CK2. Pharmacological inhibition of CK2 restores those neurons' firing peripheries. This has the potential to bridge the research between molecular level and behavior phenotype.

Comments:

On behavior:

The claim that the demonstrated learning deficiency is a result of reinforce learning needs more analysis. Furthermore, behavior experiments are somewhat dissociated with slice experiments.

As shown in Fig 1d and 1e, there are no significant difference in terms of number of earned pellets at FR30 (Fig 1d) or breakpoint at PR session (Fig 1e). While it is novel to evaluate AD mouse model with reward-learning paradigm, there could be many factors (such as motor deficits as noted by authors) that could lead to the different behavior outcomes

between 3xTg group and WT group. For instance, the 12-month 3xTg groups mice seem to have much smaller number of correct side nose poke than other groups (Fig 1c). This lack of exploring activity could lead to learning deficiency. To rule out this factor, it would be informative to plot the (binned) time course of number of noise poke.

To address this concern, we have now conducted a separate locomotor study in 3- and 12-month-old mice and show the results in Fig. 1. Consistent with literature, we actually detected an increase in locomotion and time in center of an open field in 3xTg mice. This suggests that if anything 3xTg mice exhibit more exploratory behavior than controls in a novel environment, and this does not explain a decrease in operant learning in 3xTg mice.

3xTg mice have sex-specific deficit (Juvenile 2021 Front Neurosci; cited by authors also). Are there learning differences between female and male mice in each group?

3xTg-AD mice indeed have a long history of phenotypic variation not only between sexes, but across labs. Our group has always performed all experiments in the 3xTg mice on both sexes and have never detected a sex effect, despite searching for one. We use male and female mice here in all studies, including operant self-administration behavioral paradigms. We now note this in the Methods section.

Is the recorded brain slice from mice that underwent operant conditioning task? It would be interesting to sort 3xTg mice according to their behavior outcome (learner or non-learner) and perform the subsequent electrophysiology and/or sequencing experiment. For example, the fact that 18-month groups do not show significant differences (Fig2 b) further increases the concern that variability of 3xTg groups.

We deliberately did not record from mice that underwent behavioral assays. It has been shown that both feeding state and operant self-administration of food reward can affect either dopamine neuron firing or excitatory neurotransmission in the VTA (Marinelli et al., 2006, PMID 16613555; Chen et al 2008, PMID 18667156; Branch et al., 2013, PMID 23966705; Godfrey & Borgland, 2020, 32886798). Thus, we chose to avoid this potential confound. We have added Discussion concerning the lack of certain effects at the 18-month time point (Lines 338-350). We suggest that age-dependent changes in dopamine neuron function, which we previously reported in 18-month-old wild type mice (Howell et al., 2020, PMID 32846275) may be in part masking the effect. We do not suspect a survivorship bias as we did not detect a significant change in the overall TH positive neuron numbers (Supp. Fig. 7).

Fig 1 panel 1A and especially the diagram of FR5 to PR is not very informative. It should either be removed or a pointer to the operant conditioning behavior method section should be added.

We have restructured our behavioral schematics in Fig. 1 and Supp. Fig. 5. to be more informative.

On electrophysiological recordings:

It is applauded that authors provide source code of analysis. However, the code is poorly organized: all functions are grouped in a single code block without logical order and lack of comments. There is no single example to show how the various functions were used. For example, it appears that authors have two versions of functions to perform feature analysis, one used the eel (from Blue Brain project; <https://efel.readthedocs.io/en/latest/>) and other custom written (as authors stated in the method section). Without actual examples, it is hard to assess which version the author used to obtain the results reported in the draft. Nonetheless, the criterion for $dev./dt$ in both versions of code is 20 (“`efel.api.setDerivativeThreshold(20)`” for eFEL, “`dvd_t_threshold=20`” for custom code), rather than 10 mV/ms as authors written in the method section. It is possible those values are just placeholder of parameters that are overwritten when the functions are invoked. Making this clear by providing some examples would be helpful.

We sincerely appreciate the investigation into our code repository. We agree that the code was not user friendly and contained duplicated functions. We have now revamped the GitHub page associated with the manuscript (https://github.com/heblanke/CK2_SK_ephys_and_image_analysis). The repository now has folders that contain simplified Jupyter notebooks that correspond to specific analyses along with raw data files and sample images to run example analyses. It was a correct assumption that “`dvd_t_threshold=20`” was a placeholder for manipulation, while all data presented in the manuscript using the eFEL pipeline (only Fig. 2e and 2f) utilize $dVdt=10$.

Examples of these analyses are now housed under the “BlueBrainProject_Implementation” folder. We would like to note that all spontaneous firing and whole cell voltage clamp analysis was performed with custom written scripts, examples of which are listed in the “spontaneous_firing_analysis” and “whole_cell_voltage_clamp_analysis” folders, respectively, under the main repository. The repository also houses an updated “funcs_for_ck2_sk_study.py” file that contains all functions used in the manuscript. These functions have been annotated. While specific examples are not included for each and every function, they are easily adapted within the example Jupyter notebooks for appropriate raw (AxoGraph) files.

For the perforated patch recoding data: is the apparent spike threshold (Figure 2j) and spike width calculated from all spikes from all current steps with spikes? Giving long stimulation time (1s), it is worth checking if those results hold when restricted the analysis to spikes to near rheobase and first spikes of each current step to rule out other factors such as spike adaptation.

We apologize for the lack of clarity, as we agree that spike adaptation could alter both measurements across a one second long depolarization step. Fig. 2g-2m are exclusively from perforated patch spontaneous firing, not current injection (this is also the case in Fig. 3g). The calculation for spike threshold and spike width were indeed collected by averaging the mean spike waveform from a single neuron during spontaneous firing activity in the perforated patch configuration. We have added text to the manuscript to clarify this (Lines 109-110).

The title (“Enhancing SK function normalized firing of 3xTg DA neurons”) for the NS309 results is misleading, since all the measured features for 3xTg neurons are not significant. A direct summary of decreased sensitivity to modulation by NS309 would be more appropriate.

We apologize for this oversight. We agree that there is minimal effect of NS309 in 3xTg DA neurons and this has now been remedied.

On patch-seq results:

Patch-seq experiments have very interesting results and are the highlight of this paper. Unfortunately, the number of samples are in the low end (there are a total of 34 sequenced cells from two groups. How many cells are there in each group?). The criterion to filter transcripts (“average read count ≥ 1 in at least 30% of cells in at least one group”) is generous giving the number of samples. It would be helpful to provide further information about quality control metrics (See Fig 2 of Luecken MD, Theis FJ. Current best practices in single-cell RNA-seq analysis: a tutorial. Mol Syst Biol. 2019).

To address this concern, we have now replicated the entire Patch-Seq experiment with a larger sample size and now only report the new data (Fig. 5). We have approximately doubled the number of cells from the initial submission (35 WT and 33 3xTg cells), and the new experiment confirms our initial findings. This figure now includes better verification of cell identity, and we have also added additional insight into sequencing, cell, and expression metrics (Supp. Fig. 9).

Fig 5b. Additional figures (such as scatter plot) to show the relationship between amplitude of tail current AUC and amplitude of 1st PC would be helpful.

New Fig. 5e now shows a plot of differential gene expression foldchange (3xTg to WT) versus Euclidean correlation to tail current peak amplitude. The raw data for this panel is also available in Supplemental Data Table 1.

The fact that the majority of DEG (99.59%; 14 out of 3387) are upregulated in WT group suggests there may be technical noises other than the actual biological difference. Again, increasing the number of samples and criteria of QC may overcome this.

In the new Patch-seq dataset, the distribution is less skewed with upregulated and downregulated DEGs are more balanced than previously. We would add that we have now performed a total of four Patch-seq experiments in different neuron types in 3xTg mice and in each case the number of up-regulated genes has far exceeded the down-regulated genes. Additionally, we have included a QC figure as Supplemental Figure 9.

Figure 5d. The method to calculate the correlation between amplitude of tai current AUC and gene counts of DEGs needs to be further clarified. The text only mentioned “|Euclidian| 0.6 to 1.0.” My understanding is that for each DEG, there are a pair (number of reads for the gene, amplitude of tail current AUC) of N (N = number of cells; 34 in this case) values. Then correlation is calculated between these N pairs using averaged absolute Euclidian distance? Is there a normalization step somewhere to clip the value below 1? This procedure is carried out independently for all 3387 DEGs.

We have now revised our presentation of the correlation to tail current amplitude (Figure 5E) with the new Patch-seq study. The Euclidean distance was calculated from the normalized log expression values paired to the tail current amplitude measured in that same cell. This data follows a normal distribution as is needed for Euclidean distance measures. This was then plotted against fold change between 3xTg and WT mice to demonstrate that downregulated genes in 3xTg vs WT are associated with increased tail current amplitude. The genes upregulated in 3xTg cells demonstrate a range of associations with tail current amplitudes.

While Kcnn3 expression (not shown in manuscripts) shows no significant changes between WT and 3xTg groups, It is interesting that Csnk2a1 (gene coding for CK2) is among the 31 up-regulated DEGs that are negatively correlated with tail current AUC. However, what about the 111 upregulated and 14 downregulated DEGs that are also positively correlated with tai current AUC? Adding a supplementary table to list all these genes and their functional descriptions would be extremely helpful.

We have now included protein level analysis of SK3 (Fig. 6f,g). Additionally, we have included a complete list of all differentially expressed genes and their Euclidean correlation to tail current max amplitude in Supplemental Data Table 1.

On age dependent effects:

Although authors have characterized VTA DA neurons at four ages (3, 6, 12, 18 month), and later focused on 12-month age, as the latter shows the most noteworthy results. There are some scattered comments of age dependent effects in the main text. However, a more detailed discussion would be useful for readers to further assess the aged related effects found in the manuscript.

We have now enhanced our discussion of age-dependent physiological alterations. Our lab has a long-standing interest in the effects of age on the physiology of SNc dopamine neurons (Branch et al. 2014, PMID 25009264; Howell et al. 2020, PMID 32846275; Troyano-Rodriguez et al. 2023, PMID 36658269). We have added a description of disease progression that corresponds to our selected timepoints (Lines 63-67). Additionally, in the Discussion, we suggest that some, but not all the effects detected in 3xTg mice at 18 months may be influenced by other factors such as age, survivorship bias, or homeostatic mechanisms.

Minor:

Figure 3a: the time of voltage traces are not aligned with the current step.

This has been corrected.

Figure 5d: this Venn diagram is informative, but confusing. Would be helpful to highlight the regions for the regions that have numbers 14, 31, 111 with more distinctive color and additional legend. Or put an upward arrow in the region for region 31 and downward arrow for region 14.

In our new Patch-seq figure (Fig. 5) we have completely removed the Venn diagram and replaced it with more informative metrics, such as correlations between single cells and gene expression level.

Line 400: what are the seven different measured features?

We have also replaced the PCA with a statistics table indicating the differences in firing rate among different conditions (Supp. Table 3).

Extended Figure 5c is too cluttered. Why not separate groups into different subpanels by manipulation type? For example, 3XTG vs 3xTG + SGC in one panel (maybe WT as light background). NS309 as a separate panel. Beside this, the

segregation between group (3XTg, WT + apamin) and group (WT, 3XTG + SGC) are questionable. It is better to have quantitative statistics to back up the claim.

We agree. We have now replaced the PCA with two statistical tables which include all electrophysiological differences between WT and 3xTg DA neurons (Supplemental Table 2) and effects of pharmacological manipulations to WT and 3xTg DA neurons (Supplemental Table 3).

We thank reviewers for their careful reading of our manuscript. One reviewer shared some lingering minor issues. We have included some additional analysis below and made minor text additions to the manuscript to increase clarity.

Reviewer #3 (Remarks to the Author):

Thank you for conducting the additional locomotor studies. The results convincingly demonstrate that 3xTg mice have increased rather than decreased locomotion. I have two comments to add:

1). When comparing the residency plots between WT and 3xTg, the center regions of WT mice seem more uniform (with values close to zero). This suggests that 3xTg mice tend to avoid the center, which is particularly evident in the “dark” region in Fig 1e compared to 1d. This observation contrasts with the results of Figure 1i and the authors' conclusion that “3xTg mice also spent significantly more time in the center of the locomotor area.”

I'm sorry that we were unclear. We have now updated the residency plots in three ways to better address this. First, in addition to log-transforming the data as before, we have now rescaled the data between minimum and maximum points so that the scale is more reader friendly. Second, we added a white, dashed box separating the inner and outer domains of the locomotor arena. Finally, we changed the colors for the heatmap from viridis to turbo, a palette designed by Google research, to better display the areas of high and low residence. We believe these three changes dramatically increase the readability of Figure 1. One additional clarifying point is that the tornado and residency plots are representative (i.e. one animal) and do not reflect the total heterogeneity displayed by the mice, which we have now made more clear.

2). The tornado plots seem to show that WT mice's activity decreased over time, likely due to exploration and habituation with the chamber. However, 3xTg mice's activity appears uniform over time. Quantifying the dispersion over time could serve as a behavioral metric to differentiate the phenotype.

We agree that the 3xTg representative tornado plots do display this characteristic. This is likely due to tornado plots only representing the first thirty minutes of behavior. To further quantify the locomotor data, we here have plotted five-minute bins of locomotor activity for the entirety of the session (sixty minutes, Reviewer Figure 1). We then fit a line to the behavior of each individual animal and reported the slope. The data indicate that 3xTg mice in fact decrease their activity more over the course of the session than WT mice (indicated by a steeper negative slope, Figure 2b). This is potentially an artifact of a greater starting value; however, these data align with previous locomotor reports in 3xTg mice^{1,2}. We did not include these data in the manuscript.

Overall, these results align with the cognitive impairment (there is no reward associated here) observed in 3xTg mice.

Thank you for the organization of the code in your GitHub repo and for providing better annotation and examples. Since all the functions are contained in the “funcs_for_ck2_sk_study.py” file, it is not necessary to redefine functions in the notebooks (instead, import only those needed from the .py file). This would help reduce errors due to replication and allow readers to focus more on the analysis pipeline. If figures 2e and 2f are using the eFEL pipeline instead of a custom script, mentioning that in the corresponding method section would be appropriate.

Reviewer Figure 1. 3xTg mice exhibit more locomotion in an open field across time than age-matched WT mice. a. Average distance in 5-minute bins across the entirety of a 60-minute open field session shows 3- and 12-month-old 3xTg mice show more forward locomotion than WT mice of the same age with a stereotypic decrease in locomotor activity present in both genotypes at both ages. **b.** To represent locomotor change over the session, a linear regression was fit to distance traveled by time. There was a significant genotype effect (One way ANOVA, $F_{1,34} = 4.757$, $P = 0.0362$) but no significant effect of age (One way ANOVA, $F_{1,34} = 0.6604$, $P = 0.4421$). Furthermore, there was no specific effect between genotype at either 3 mo (Sidak's, $P = 0.2625$) or 12 mo (Sidak's, $P = 0.2312$). Error bars represent SEM.

We have updated the repository examples to call functions from the .py file, and the .py file has been added to each individual example folder so it maintains its self-contained nature. The use of eFEL in figures 2e and 2f has been included in the methods.

The quality and quantity of patch-seq data, as well as the associated analyses, have been greatly improved. A few comments:

1). Lines 234 to 236: "4,165 genes passed the criteria (read count >1 in >80% of samples from at least one group) for expression and transcriptome profiles, and WT and 3xTg cells generally separated by t-distributed stochastic neighbor embedding (t-SNE; Figure 5c)." I suggest splitting this sentence into two. The second half, "WT and 3xTg cells generally separated by t-distributed stochastic neighbor embedding (t-SNE; Figure 5c)," is poorly written and not accurate. The two groups are not generally (linearly) separable in the t-SNE map. However, based on your DEG analysis, a generic classifier in the original (4165) dimension would give you good classification.

We agree that the t-SNE results were over-interpreted. We have corrected this in line 239.

2). Please add a formal description of how you are calculating "Euclidean correlation" in the method section. If it is standard Euclidean distance, it should not be between -1 and 1, regardless of normalization. Please provide justification for why it is preferred over standard Pearson correlation (or demonstrate that the latter provides comparable results). Please replace "Euclidean similarity" (in Fig 5d caption and supp. Table 1) with "Euclidean distance" if they refer to the same metric.

Thank you for catching this. Our Euclidian graph relied on Strand NGS software that we now believe might have a flawed algorithm. We have instead now computed Pearson correlations in place of all Euclidean distance calculations (Figure 5e, supplemental table 1). While we observed a shift in datapoints, the overall conclusion from the panel remains the same.

3). There is a recent large single-nucleus sequencing dataset of midbrain DA neurons in mice publicly available (Salmani et al. (2024), "Transcriptomic atlas of midbrain dopamine neurons uncovers differential vulnerability in a Parkinsonism lesion model," eLife 12). Integrating an analysis between your data and this dataset could provide further insights. For example, you could check if VTA region-specific Calb1 (Figure 3B in Salmani 2024) is expressed in most patch-seq data, or if patch-seq VTA data are uniformly distributed among the reference single-nucleus data (by "mapping" the former onto the latter) to check for sampling bias. In a future study, you might consider using a similar approach to compare VTA DA neurons in 3xTg mice and a control group using high-throughput single-nucleus sequencing, as VTA-specific midbrain DA neurons are readily separable from the rest. This could be particularly relevant when comparing different age groups, assuming the regional differences hold across ages.

Thank you for the suggestion. We agree that interpreting our results with this manuscript and others like it as they become available has value, and will do so in future studies. For now, we have added a reference to Salmani et al., 2024 (line 329) and a brief discussion about potential roles for calbindin in neuronal vulnerability in AD and PD models. Calb1 was detected in 64 out of 68 of our sequenced neurons, suggesting that a typical population of VTA cells was sampled. Further, Calb1 was upregulated 3.76 fold in 3xTg mice, which has previously been cited as a potential neuroprotective adaptation. We also added this gene to our identification matrix in Figure 5B.

Reviewer #3 (Remarks on code availability):

Results in paper are not reproducible from the current status of the code and associated instructions.

The code is a usable resource, although a better refactoring would make it more user-friendly. README file doesn't include how to run the application, but it should be easy to install and run the code.

I did not attempt to install or run the code.

We have updated the README page. The code contained in the mentioned repository does not initiate an application, or graphical user interface, but is to be run as either raw python code or in a Jupyter

notebook. Currently, the repository contains examples of data, and the code is heavily annotated. We have also included in the README file a link to GitHub's instructions on downloading a repository and subsequent instruction on how to run the example notebooks.

REFERENCES

- 1 Gloria, Y., Ceyzeriat, K., Tsartsalis, S., Millet, P. & Tournier, B. B. Dopaminergic dysfunction in the 3xTg-AD mice model of Alzheimer's disease. *Sci Rep* **11**, 19412 (2021). <https://doi.org/10.1038/s41598-021-99025-1>
- 2 Pietropaolo, S., Feldon, J. & Yee, B. K. Age-dependent phenotypic characteristics of a triple transgenic mouse model of Alzheimer disease. *Behav Neurosci* **122**, 733-747 (2008). <https://doi.org/10.1037/a0012520>

There were no remaining reviewer concerns to address. We thank the editors and reviewers for their thoughtful comments throughout the submission process.

This manuscript aims to identify pathological mechanisms in single VTA DA neurons of AD mice model (3xTg) at two ages, using patch-clamp recording, single-cell sequencing and behavior test. Results show DA neurons in disease model have hyperexcitability associated with decrease of SK channel currents which in turn was affected by upregulation of CK2. Pharmacological inhibition of CK2 restores those neurons' firing peripheries. This has the potential to bridge the research between molecular level and behavior phenotype.

Comments:

On behavior:

The claim that the demonstrated learning deficiency is a result of reinforce learning needs more analysis. Furthermore, behavior experiments are somewhat dissociated with slice experiments.

- 1) As shown in Fig 1d and 1e, there are no significant difference in terms of number of earned pellets at FR30 (Fig 1d) or breakpoint at PR session (Fig 1e). While it is novel to evaluate AD mouse model with reward-learning paradigm, there could be many factors (such as motor deficits as noted by authors) that could lead to the different behavior outcomes between 3xTg group and WT group. For instance, the 12-month 3xTg groups mice seem to have much smaller number of correct side nose poke than other groups (Fig 1c). This lack of exploring activity could lead to learning deficiency. To rule out this factor, it would be informative to plot the (binned) time course of number of nose poke.
- 2) 3XTg mice have sex-specific deficit (Juvenile 2021 Front Neurosci; cited by authors also). Are there learning differences between female and male mice in each group?
- 3) Is the recorded brain slice from mice that underwent operant conditioning task? It would be interesting to sort 3XTg mice according to their behavior outcome (learner or non-learner) and perform the subsequent electrophysiology and/or sequencing experiment. For example, the fact that 18-month groups do not show significant differences (Fig2 b) further increases the concern that variability of 3xTg groups.
- 4) Fig 1 panel 1A and especially the diagram of FR5 to PR is not very informative. It should either be removed or a pointer to the operant conditioning behavior method section should be added.

On electrophysiological recordings:

It is applauded that authors provide source code of analysis. However, the code is poorly organized: all functions are grouped in a single code block without logical order

and lack of comments. There is no single example to show how the various functions were used. For example, It appears that authors have two versions of functions to perform feature analysis, one used the eel (from Blue Brain project; <https://efel.readthedocs.io/en/latest/>) and other custom written (as authors stated in the method section). Without actual examples, it is hard to assess which version the author used to obtain the results reported in the draft. Nonetheless, the criterion for dev./dt in both versions of code is 20 (“efel.api.setDerivativeThreshold(20) “ for eFEL, “dvd_t_threshold=20” for custom code), rather than 10 mV/ms as authors written in the method section. It is possible those values are just placeholder of parameters that are overwritten when the functions are invoked. Making this clear by providing some examples would be helpful.

For the perforated patch recoding data: is the apparent spike threshold (Figure 2j) and spike width calculated from all spikes from all current steps with spikes? Giving long stimulation time (1s), it is worth checking if those results hold when restricted the analysis to spikes to near rheobase and first spikes of each current step to rule out other factors such as spike adaptation.

The title (“Enhancing SK function normalized firing of 3xTg DA neurons”) for the NS309 results is misleading, since all the measured features for 3xTg neurons are not significant. A direct summary of decreased sensitivity to modulation by NS309 would be more appropriate.

On patch-seq results:

Patch-seq experiments have very interesting results and are the highlight of this paper. Unfortunately, the number of samples are in the low end (there are a total of 34 sequenced cells from two groups. How many cells are there in each group?). The criterial to filter transcripts (“average read count ≥ 1 in at least 30% of cells in at least one group”) is generous giving the number of samples. It would be helpful to provide further information about quality control metrics (See Fig 2 of Luecken MD, Theis FJ. Current best practices in single-cell RNA-seq analysis: a tutorial. Mol Syst Biol. 2019).

- 1) Fig 5b. Additional figures (such as scatter plot) to show the relationship between amplitude of tail current AUC and amplitude of 1st PC would be helpful.
- 2) The fact that the majority of DEG (99.59%; 14 out of 3387) are upregulated in WT group suggests there may be technical noises other than the actual biological difference. Again, increasing the number of samples and criteria of QC may overcome this.
- 3) Figure 5d. The method to calculate the correlation between amplitude of tail current AUC and gene counts of DEGs needs to be further clarified. The text only mentioned “|Euclidian| 0.6 to 1.0.” My understanding is that for each DEG, there are a pair (number of reads for the gene, amplitude of tail current AUC) of N ($N =$

number of cells; 34 in this case) values. Then correlation is calculated between these N pairs using averaged absolute Euclidian distance? Is there a normalization step somewhere to clip the value below 1? This procedure is carried out independently for all 3387 DEGs.

- 4) While Kcnn3 expression (not shown in manuscripts) shows no significant changes between WT and 3xTg groups, It is interesting that Csnk2a1 (gene coding for CK2) is among the 31 up-regulated DEGs that are negatively correlated with tail current AUC. However, what about the 111 upregulated and 14 downregulated DEGs that are also positively correlated with tai current AUC? Adding a supplementary table to list all these genes and their functional descriptions would be extremely helpful.

On age dependent effects:

Although authors have characterized VTA DA neurons at four ages (3, 6, 12, 18 month), and later focused on 12-month age, as the latter shows the most noteworthy results. There are some scattered comments of age dependent effects in the main text. However, a more detailed discussion would be useful for readers to further assess the aged related effects found in the manuscript.

Minor:

Figure 3a: the time of voltage traces are not aligned with the current step.

Figure 5d: this Venn diagram is informative, but confusing. Would be helpful to highlight the regions for the regions that have numbers 14, 31, 111 with more distinctive color and additional legend. Or put an upward arrow in the region for region 31 and downward arrow for region 14.

Line 400: what are the seven different measured features?

Extended Figure 5c is too cluttered. Why not separate groups into different subpanels by manipulation type? For example, 3XTG vs 3XTG + SGC in one panel (maybe WT as light background). NS309 as a separate panel. Beside this, the segregation between group (3XTg, WT + apamin) and group (WT, 3XTG + SGC) are questionable. It is better to have quantitative statistics to back up the claim.